



# Decomposing the response of the stratospheric Brewer-Dobson circulation to an abrupt quadrupling in $CO_2$

Andreas Chrysanthou[1], Amanda C. Maycock[1] and Martyn P. Chipperfield[1]

[1] School of Earth and Environment, University of Leeds, Leeds, LS2 9JT, UK

**Correspondence**: Andreas Chrysanthou (eeac@leeds.ac.uk)

**Abstract.** We perform 50-year-long time-slice experiments using HadGEM3-A to decompose the long-term response of the Brewer-Dobson circulation (BDC) to an abrupt quadrupling in $CO_2$ ($4\times CO_2$) into: 1) a rapid atmospheric adjustment; 2) a contribution from the global-average sea surface temperature (SST) change (+3.4 K); and 3) an SST pattern effect. The SST fields are derived from the CMIP5 multi-model ensemble. Two further experiments explore the impact on the BDC of the spread in global-average SST response to $4\times CO_2$ across the CMIP5 models (range 2.1-4.9 K). At 70 hPa (10 hPa) the annual mean tropical upward mass flux increases by 45% (35%) due to the $4xCO_2$ perturbation. At 70 hPa, around 70% of the increase is from the global-uniform SST warming, with the remainder coming in similar contributions from the rapid adjustment and SST pattern effect. In contrast, at 10 hPa the total mass flux increases by 35% and comes mainly from the rapid adjustment (~40%) and the uniform SST warming (~45%), with a small contribution from the SST pattern. Therefore, at 10 hPa the magnitude of the spread in global uniform SST response is comparable to the rapid adjustment. Conversely, at 70 hPa the effect of spread in global-mean SST is larger than both the rapid adjustment and the SST pattern effect. We derive an approximately linear sensitivity of the tropical upward mass flux to global surface air temperature change of $0.62 \times 10^9$ kg $s^{-1}$ $K^{-1}$ (9% $K^{-1}$) at 70 hPa and $0.10 \times 10^9$ kg $s^{-1}$ $K^{-1}$ (6% $K^{-1}$) at 10 hPa. The results confirm the most important factor for the acceleration of the BDC in the lower stratosphere under increased $CO_2$ is global SST change. We also quantify for the first time that the rapid adjustment to $CO_2$ is of similar importance to SSTs for the increased BDC in the upper stratosphere. This demonstrates a potential for a fast and slow timescale of the response of the BDC to greenhouse gas forcing, with the relative prominence of those timescales being height dependent.

## 1 Introduction

The residual circulation in the stratosphere, or the Brewer–Dobson circulation (BDC), is characterised by slow ascent in the tropics, poleward flow and downwelling in the subtropics and extratropics (Andrews et al., 1987; Holton et al., 1995; Plumb, 2002). There is a strong seasonality in the strength and width of the BDC (Rosenlof, 1995). In the winter hemisphere, the poleward mass transport that occurs in the middle and upper stratosphere is termed the deep branch, while the shallow branch in the lower stratosphere is present year round in both hemispheres (Birner and Bönisch, 2011). The BDC controls the transport and distribution of radiatively active trace gases such as stratospheric ozone and water vapour (Brewer, 1949; Dobson, 1956), as well as the lifetimes of chemically-important trace gases such as chlorofluorocarbons (CFCs; Butchart and





Scaife, 2001). The BDC is a wave-driven circulation forced by breaking of planetary-scale Rossby waves and small-scale gravity waves (Holton et al., 1995). The torque imposed by the wave breaking allows flow across lines of constant angular momentum.

General circulation models (GCMs) and chemistry-climate models (CCMs) consistently simulate an acceleration of the BDC in scenarios that include increasing greenhouse gas concentrations (Rind et al., 1990, 2002; Sigmond et al., 2004; Butchart et al., 2006, 2010; Fomichev et al., 2007; Olsen et al., 2007; Deckert and Dameris, 2008; Garcia and Randel, 2008; Li et al., 2008; Calvo and Garcia, 2009; McLandress and Shepherd, 2009; Oman et al., 2009; SPARC, 2010; Garny et al., 2011; Shepherd and McLandress, 2011; Lin and Fu, 2013; Hardiman et al., 2014). A strengthened BDC increases

stratosphere-troposphere exchange (STE) of ozone (Rind et al., 2001; Hegglin and Shepherd, 2009; Banerjee et al., 2016) and affects projected ozone trends in the tropical lower stratosphere (e.g. Keeble et al., 2017), subtropics (e.g. Li et al., 2009), and polar regions (e.g. Oman et al., 2009).

The wave forcing that drives the BDC arises from various types of waves generated in the troposphere with different temporal and spatial scales (e.g., Randel et al., 2008), which propagate upwards, break and dissipate their momentum and

energy (Holton et al., 1995; Plumb and Eluszkiewicz, 1999; Semeniuk and Shepherd, 2001). Changes in the BDC must, therefore, be accompanied by changes in stratospheric wave forcing. Three main mechanisms for the altered wave forcing of the BDC under climate change have been considered in the literature: 1) changes in the strength of tropospheric wave generation; 2) changes in the latitudinal distribution of wave forcing within the stratosphere in the vicinity of the turnaround latitudes, where the residual vertical velocity changes sign; and 3) changes in the vertical penetration of tropospheric wave

forcing into the stratosphere. Anomalous wave activity emanating from the extratropical troposphere has been shown to have a minimal impact on the overall strength of the BDC (Butchart and Scaife, 2001; Sigmond et al., 2004; Garcia and Randel, 2008; Garny et al., 2011) owing to the fact the wave forcing needs to occur in the vicinity of the turnaround latitudes.

Many studies have pointed to an important role for the projected strengthening and upward shift of the subtropical jets with tropospheric warming to explain the modelled increase in the BDC under climate change. This robust change in the

pattern of zonal winds (Collins et al., 2013) moves the Rossby wave critical layers in the lower stratosphere upward (Randel and Held, 1991), enabling enhanced penetration of Rossby wave activity in the subtropical lower stratosphere and an altered distribution of momentum deposition (Rind et al., 1990; Garcia and Randel, 2008; McLandress and Shepherd, 2009; Calvo and Garcia, 2009; Garny et al., 2011). In support of this theoretical basis, the multi-model spread in the end of 21[st] century lower stratospheric zonal wind trends near the turnaround latitudes was found to explain ~70% of the spread in tropical

upward mass flux trends in the lower stratosphere across a set of CCMs (Lin and Fu, 2013). Some studies have also found a role for enhanced excitation of tropical waves (equatorially trapped quasi-stationary Rossby waves) under climate change for a strengthened BDC (Deckert and Dameris, 2008; Calvo and Garcia, 2009), but the potential for this to drive an increase in the total mass circulation rather than simply a redistribution within the tropics has been questioned (Garny et al., 2011; Shepherd and McLandress, 2011).



Although the signal of an increased BDC in a warmer climate is a highly robust feature of GCMs and CCMs, there are differences amongst models in the relative contributions to the increase from resolved and parameterized wave forcing (Butchart et al., 2006, 2010; Garcia and Randel, 2008; Calvo and Garcia, 2009; McLandress and Shepherd, 2009; Garny et al., 2011). This may be related to models having different climatological resolved and parameterized wave forcing (e.g., Chrysanthou et al., 2019) and the potential for a compensation effect between the different types of wave forcing in driving a

change in the BDC (e.g. Cohen et al., 2014; Sigmond and Shepherd, 2014).

To understand the relative importance of different drivers, some modelling studies have performed idealised experiments to decompose the BDC response to climate change into different components. Sigmond et al. (2004) performed experiments with the Canadian Middle Atmosphere Model (CMAM) in which $CO_2$ was doubled separately in the troposphere and stratosphere. In each case, sea surface temperature (SST) changes were imposed as a fraction of the total SST response

according to their respective radiative forcings. Sigmond et al. (2004) showed that the increase in residual circulation in DJF caused a small warming in the Arctic lower stratosphere, of which about two thirds could be attributed to the tropospheric $CO_2$ doubling and about one third to the middle-atmospheric $CO_2$ doubling. Their results were qualitatively consistent with the seminal results of Rind et al. (1990) who performed comparable experiments with the NASA Goddard Institute for Space Studies (GISS) model but over a shorter period.

Olsen et al. (2007) performed experiments for the period 1949 to 1998 with the NASA GEOS-4 GCM using prescribed observed SSTs. They attributed the increase in residual circulation between the first and last decades of their simulations to a stronger SST gradient between the tropics and middle latitudes, resulting in a greater meridional temperature gradient in the subtropical troposphere and more poleward refraction of planetary-scale Rossby waves in the lower stratosphere. Further simulations by Olsen et al. (2007) added the radiative effects of atmospheric GHG changes and showed a small but

insignificant increase in STE trend compared to the SST-only experiments. Oman et al. (2009) performed sensitivity experiments with the GEOS-CCM model (based on GEOS-4) to decompose the relative effects of SSTs, GHGs and halogens on the stratospheric age of air distribution between 1960 and 2100. To isolate the effects of SST changes, they compared simulations using SSTs from two different climate models that differed in their climatological SST. They describe the SST experiment as "tropical SSTs" though the SST changes appear to be imposed globally. This comparison further combines the

effects of differences in both global mean SST and SST patterns between the two climate model datasets, though this was not explicitly discussed. As with all other similar studies, they concluded that increased SSTs contribute to an increase in tropical lower stratospheric upwelling and a decrease in age of air.

While studies have demonstrated that warmer SSTs increase the strength of the BDC (Olsen et al., 2007; Oman et al., 2009; Lin et al., 2015), one confounding factor in the literature is that the SST response to climate change in the tropical

Pacific often shows an El Nino-like pattern (Latif and Keenlyside, 2009). El Niño Southern Oscillation (ENSO) itself affects the BDC both through modulation of the Northern hemisphere (NH) winter stratospheric circulation (Manzini et al., 2006) and tropical lower stratospheric upwelling (Marsh and Garcia, 2007; Randel et al., 2009). Using CMAM, Simpson et al. (2011) attribute the increase in boreal winter tropical lower stratospheric upwelling under El Niño to increased resolved



wave forcing in the Southern hemisphere (SH) subtropical lower stratosphere, which was caused by altered wave sources in
the troposphere under El Niño. In contrast, Calvo et al., (2010) using Whole Atmosphere Community Climate Model
(WACCM), attribute the increased tropical upwelling during El Niño to changes in the propagation and dissipation of
parameterized gravity waves caused by the anomalous location and intensity of the subtropical jets highlighting the model-
specific nature of gravity wave drag parameterizations. A uniform SST increase can generate most of the canonical pattern of
long-term tropical upper tropospheric warming, through impacts on tropical convection and the water vapour and lapse rate
feedbacks (e.g. Chen et al., 2013). Lin et al. (2015) showed an approximately linear relationship between tropical annual
mean surface temperature and anomalous lower stratospheric mass flux in the GFDL-CM3 model that held on interannual
(i.e. ENSO), decadal and centennial timescales. However, on multi-decadal timescales this calculation aliases the direct
atmospheric radiative effects of GHGs, the SST pattern effect and the SST magnitude into one term. Hence, under climate
change the direct radiative effect of $CO_2$ (rapid adjustment) (Sigmond et al., 2004; Olsen et al., 2007), as well as the
magnitude and pattern of SST change may contribute to simulated changes in the BDC.

While previous literature suggests that the three distinct effects may contribute to projected changes in the BDC, no
previous study has explicitly quantified their importance; this is the goal of our study. Here, we perform climate model
experiments to decompose the response of the BDC to an abrupt quadrupling of $CO_2$ into three components: 1) the rapid
adjustment, or direct component, associated with $CO_2$ radiative effects in the absence of SST change; 2) a global uniform
SST warming; and 3) the SST pattern effect. The goal is to understand the distinct contributions of the three components and
assess the extent to which they can be combined to explain the overall BDC response. We further compare the magnitudes of
the rapid adjustment and SST pattern effects on the BDC with the effect of spread in global warming due to $CO_2$ across
climate models. The remainder of the paper is laid out as follows: Section 2 describes the atmospheric model and
experimental set-up; Section 3 presents the results and Section 4 summarises our main findings and conclusions.

## 2 Data and Methods

### 2.1 Atmospheric model description

We use the Hadley Centre Global Environment Model version 3 (HadGEM3) variant of the Met Office Unified Model
(MetUM) version 8.4, which has been used for both numerical weather prediction and climate simulation. It is configured
with the Global Atmosphere (GA4.0) and comprises a non-hydrostatic fully compressible dynamical core that uses a semi-
implicit semi-Lagrangian advection scheme in terrain-following hybrid height coordinates (Walters et al., 2014). We run the
atmosphere-only configuration (HadGEM3-A) at N96 horizontal resolution (1.875° x 1.25°, ~135 km in mid-latitudes) with
85 levels (L85) from the surface to an altitude of ~85 km. Interactions of the flow blocking drag associated with the
orographic gravity wave drag (OGWD) are parameterized, as detailed in Webster et al. (2003). Similarly, a spectral sub-grid
parameterization is used for the representation of the gravity wave drag induced in the upper stratosphere and mesosphere,



forced by non-orographic sources (NOGWD) such as convective processes and fronts, which enables HadGEM3 to simulate a realistic quasi-biennial oscillation (QBO) as detailed in Scaife et al. (2002).

## 2.2 Experiment design

Seven 50-year-long time-slice simulations were performed with HadGEM3-A with fixed boundary conditions including prescribed SSTs and sea ice. The experiment names and IDs are shown in Table 1. The reference simulation (run A) uses

boundary conditions, including greenhouse gas (GHG) concentrations, natural and anthropogenic primary aerosol or reactive gas emissions, set to pre-industrial (year 1850) values following the Coupled Model Intercomparison Project 5 protocol (CMIP5; Lamarque et al., 2010; Taylor et al., 2012). The reference SSTs and sea ice concentrations (SIC) are annually repeating fields taken as the monthly-mean multi-model mean (MMM) from the CMIP5 piControl simulations (Taylor et al., 2012). The MMM reference SST and SIC fields are constructed from the average of the last 150 years of the piControl

experiments from the 26 CMIP5 models listed in the supplementary material (Table S1).

Six perturbation experiments are performed to isolate different components of the long-term response in the CMIP5 abrupt-4xCO$_2$ experiment, which instantaneously quadruples CO$_2$ from its preindustrial concentration. We first calculate the CMIP5 monthly-mean MMM SST in the abrupt-4xCO$_2$ experiment using the final 50 years (years 101-150) of each model run (Table S1). The annual mean SST anomalies compared to the reference preindustrial state are shown in Figure 1(a). Note

that in all the perturbation experiments, SIC is held fixed at the reference values. This is artificial, but it enables the effects of SSTs on the BDC and associated mechanisms to be isolated from the possible effects of changing sea ice on the stratosphere, which has been proposed but is more uncertain (e.g. Kim et al., 2014; McKenna et al., 2018).

The first perturbation experiment (run B) accounts for the full (atmosphere + SST) abrupt-4xCO$_2$ response and is designed to simulate the long-term quasi-equilibrium state in response to the abrupt CO$_2$ forcing. The second perturbation

experiment (C) only accounts for the CO$_2$ rapid adjustment by quadrupling atmospheric CO$_2$ concentrations while holding SSTs and SIC at their preindustrial values. The third experiment (D) imposes a monthly-varying globally uniform SST anomaly derived from the global mean multi-model mean 4xCO$_2$ SST anomaly relative to the control. In the annual and multi-model mean this is equal to 3.4 K. In the fourth perturbation experiment (E), we subtract the monthly-varying uniform warming value from the 4xCO$_2$ anomalies to impose the local deviations in SST from the global uniform value (i.e. the SST

pattern). By design, the sum of the SSTs anomalies in runs D and E equals the full SST anomalies of run B. The annual mean SST anomalies associated with experiments B, D and E, respectively, are shown in Figure 1. Note that the change in annual mean global surface air temperature (GSAT) in runs A and C is larger than the global mean SST anomaly, partly because of an enhanced warming response over land (4.3 K and 4.0 K, respectively). While SSTs are held fixed in run B, there are changes to land temperatures that cause a small GSAT response (0.43 K). Finally, although the global mean SST

change in run D is zero by construction, there are changes to land temperatures that lead to a small GSAT response (0.45 K).

There is substantial inter-model spread in the modelled global mean SST change in the abrupt-4xCO$_2$ experiments (Flato et al., 2013). To investigate the effect of this spread on the BDC, and to place the rapid adjustment and SST pattern effects



into the context of model uncertainty in the global mean surface warming due to $CO_2$, we perform two further uniform SST warming sensitivity runs. These are chosen to be the lowest (annual mean ~2.1 K; run F) and highest (annual mean ~4.9 K; run G) global mean SST changes from the 26 CMIP5 models used in this study. These values come from the INMCM4 (Volodin, 2013) and IPSL-CM5A-LR (Dufresne et al., 2013) models, respectively. The annual mean GSAT change in runs F and G is 3.0 and 6.1 K, respectively.

Since the model does not include interactive chemistry, we prescribe preindustrial ozone concentrations in our $4xCO_2$ perturbation runs following the Coupled Model Intercomparison Project 6 (CMIP6; Eyring et al., 2016) protocol. In this case, for simplicity, we prescribe the zonal mean ozone concentrations used in the CMIP6 experiments run with HadGEM3-GC3.1 (Kuhlbrodt et al., 2018; Williams et al., 2018). It should be noted that keeping $O_3$ concentrations fixed in all our experiments will implicitly neglect the effects of any $O_3$ feedbacks from both the chemical effects of increased $CO_2$ and the transport effects from an altered BDC; this includes effects on the thermal structure of the upper troposphere especially around the tropical upper troposphere and lower stratosphere (Nowack et al., 2015; Chiodo and Polvani, 2017) and on upper stratospheric temperatures (Maycock, 2016).

## 2.3 Residual circulation diagnostics

To diagnose the BDC and its changes we use the Transformed Eulerian Mean circulation diagnostics (TEM; Andrews et al., 1987; Andrews and Mcintyre, 1976, 1978). The TEM residual circulation velocities $(\overline{v}^*, \overline{w}^*)$ are defined as (Andrews et al., 1987):

$$\overline{v}^* = -\frac{1}{\rho_0 \cdot a \cdot cos\phi}\frac{\partial \overline{\Psi}^*}{\partial z}, \quad \overline{w}^* = \frac{1}{\rho_0 \cdot a \cdot cos\phi}\frac{\partial \overline{\Psi}^*}{\partial \phi}, \qquad (1)$$

where $\rho_0$ is the log-pressure density, $a$ is the Earth radius and $\phi$ is the latitude and $\overline{\Psi}^*(\phi,z)$ is the residual meridional mass streamfunction. We further calculate the residual mass streamfunction as:

$$\overline{\Psi}^*(\phi,z) = \frac{2\pi \cdot a \cdot cos\phi}{g}\int_{bottom}^{top}\overline{v}^* \cdot dp, \quad (2)$$

where $g$ is the acceleration due to gravity. Equation 2 is integrated from the top of atmosphere to the surface using the boundary condition that $\overline{\Psi}^* = 0$ at the top of the atmosphere (p=0). Subsequently, we calculate the net downward mass flux in each hemisphere, by finding $\overline{\Psi}^*_{max}$ and $\overline{\Psi}^*_{min}$ in the NH and SH, respectively, at each pressure level. The net tropical upward mass flux, which is equal to the sum of the downward mass fluxes in each hemisphere, can then be expressed as (Rosenlof, 1995):



$$\text{Tropical upward mass flux } = 2\pi a \, (\overline{\Psi}^*_{max} - \overline{\Psi}^*_{min}) \, . \qquad (3)$$

We apply the "downward control" principle (DCP; Haynes et al., 1991) to further separate the contributions to the tropical upward mass flux from resolved waves due to the divergence of Eliassen-Palm flux (EPF) and contributions from OGWD and NOGWD. Resolved waves and parameterized gravity wave drag (OGWD/NOGWD) constitute the eddy-induced total zonal forces $\overline{F}$. Under steady-state conditions, the $\overline{\Psi}^*(\phi,z)$ at a specified log(pressure)-height, $z$, is related to the vertically-integrated $\overline{F}$ above that level along a surface of constant zonal mean absolute angular momentum $\overline{m}=a\cos\phi(\overline{v}+a\Omega\cos\phi)$, where $\overline{v}$ is the zonal mean zonal wind and $\Omega$ is Earth's rotation rate (Haynes et al., 1991). Outside of the tropics, $\overline{m}$ is approximately constant at a fixed latitude, $\phi$, resulting in the following equation (Haynes et al., 1991):

$$\overline{\Psi}^*(\phi,z) = \int_{z}^{\infty} \left\{ \frac{\rho_0 a^2 \overline{F} cos^2\phi}{\overline{m}_\phi} \right\}_{\phi=\phi(z')} , \qquad (4)$$

where $\overline{m}_\phi \approx -2\Omega a^2 \sin\phi\cos\phi$ is the quasi-geostrophic limit. The boundary conditions are $\overline{\Psi}^* \to 0$ and $\rho_0 \overline{w}^* \to 0$ as $z \to \infty$.

## 3 Results

### 3.1 Zonal mean temperature response

Figure 2 shows latitude-pressure cross-sections of the annual and zonal mean temperature anomalies from the reference preindustrial simulation for perturbation runs B, C, D and E. The full response (Fig. 2a) exhibits the canonical pattern of temperature change due to increased $CO_2$, with tropospheric warming that is maximum in the tropical upper troposphere and stratospheric cooling that increases with height (Collins et al., 2013). Note that Arctic amplification in the lower troposphere is small here compared to coupled atmosphere-ocean models (Collins et al., 2013), presumably because we do not impose sea ice changes in the runs.

The rapid adjustment due to changes in atmospheric $CO_2$ (Fig. 2b) accounts for most of the stratospheric cooling seen in the full response, with cooling of ~15 K in the upper stratosphere. However, the stratospheric cooling in run C is more uniform in latitude than in the full experiment, and more closely resembles the purely radiative response to $CO_2$ (Fels et al., 1980). In the troposphere, the rapid adjustment induces a weak (<1 K) warming that is fairly homogeneous. This weak tropospheric warming comes from heating of the land surface and changes to atmospheric absorption and emission of longwave radiation due to increased $CO_2$ concentrations. Most of the tropospheric warming is reproduced by imposing the uniform SST warming (Fig. 2c), including the tropical upper tropospheric amplification of up to 9 K and the extension of warming into the subtropical lower stratosphere in both hemispheres. In the stratosphere, the uniform SST warming induces an anomalous meridional temperature gradient, with cooling of 2-3 K in the tropical middle and upper stratosphere and





warming in the extratropics and polar regions. This pattern accounts for most of the latitudinal variation in stratospheric cooling seen in the full response (Fig. 2a).

The SST pattern experiment (Fig. 2d) exhibits a similar morphology in the temperature response to the uniform SST
warming run, albeit much weaker in amplitude. In the troposphere, the SST pattern induces a weak warming that is comparable in magnitude to the rapid adjustment (Fig. 2b), but with a weak upper tropospheric amplification that enhances the effect of the uniform warming (Fig. 2c). This upper tropospheric amplification suggests enhanced tropical deep convection, which may be consistent with the imposed anomalous tropical SST warming (Fig. 1c).

The thick yellow lines in Figure 2 denote the tropopause pressure for each perturbation experiment. These can be
compared to the climatological tropopause in the reference simulation (thick black lines). The lifting of the tropopause by ~1 km within the deep tropics in the full experiment comes mainly from the uniform SST warming (~80%) with the remaining 20% coming from the SST pattern. However, the maximum tropopause lifting (> 1.2 km) occurs in the extratropics of both hemispheres, especially over the Arctic polar cap (not shown).

## 3.2 Zonal mean zonal wind response

The annual mean zonal mean zonal wind anomalies in the four perturbation experiments are shown in Figure 3. The full $4xCO_2$ response (Fig. 3a) shows the familiar pattern of a strengthening and upward shift of the subtropical jets, a strengthening and poleward shift of the midlatitude westerlies in the SH, and increased westerlies in the SH stratosphere (Collins et al., 2013). The strengthened subtropical jets arise mainly from the uniform SST warming (Fig. 3c) with a small contribution from the SST pattern (Fig. 3d). In contrast, the rapid adjustment (Fig. 3b) does not induce a strengthening of the
subtropical jets, but it does explain a significant part of the increased westerlies in the SH extratropics, particularly in the upper stratosphere. The SST pattern effect also contributes to the increased SH stratospheric westerlies, but the uniform SST warming shows the largest increase. In the NH, the full experiment shows stronger westerlies in the lower stratosphere and weakened westerlies near the subtropical stratopause. The anomalous lower stratospheric westerlies are contributed by the uniform warming in the subtropics and midlatitudes and the rapid adjustment in the extratropics and polar region. The
uniform warming also causes weakened westerlies in the subtropical upper stratosphere, with a smaller effect from the rapid adjustment.

The full $4xCO_2$ response shows significant zonal wind anomalies in the tropical stratosphere between 50-1 hPa, which is also seen in the uniform SST warming experiment. Investigation of the QBO characteristics in the experiments reveals changes in the period, magnitude and vertical penetration of the easterly and westerly QBO phases (not shown). Related
changes to the QBO properties under climate change have been noted in other idealised GCM experiments (e.g. Kawatani et al., 2011), though a detailed investigation of the QBO is beyond the scope of this study.

The zero wind lines ($\overline{u} = 0$), which demarcate the "critical lines" for linear stationary Rossby waves (Dickinson, 1968), are shown by the thick lines in Figure 3. In the stratosphere, there is a clear equatorward contraction of the zero wind lines in both hemispheres in the full $4xCO_2$ experiment. Previous studies have connected this to increased penetration of Rossby

©️ Author(s) 2020. CC BY 4.0 License.





wave activity into the subtropical lower stratosphere (Shepherd and McLandress, 2011). The contraction of the zero wind
lines is primarily due to the uniform SST warming, with a modest contraction also seen in the rapid adjustment and SST
pattern experiments.

### 3.3 Residual circulation response

Figure 4 shows latitude-pressure cross-sections of the annual-mean residual vertical velocity ($\overline{w}^*$) anomalies with respect to
the reference simulation for experiments B, C, D and E. The uniform SST warming accounts for most of the increased
tropical lower stratospheric upwelling seen in the full $4xCO_2$ response. However, there are also significant increases in
tropical lower stratospheric upwelling induced by the rapid adjustment and the SST pattern perturbations (Fig. 4b, 4d). While
comparatively small compared to the effect of the uniform SST warming, the increases in $\overline{w}^*$ in the tropical lower
stratosphere from the SST pattern are broadly comparable in magnitude to the effects of ENSO-like SST perturbations found
in other modelling studies (Calvo et al., 2010; Simpson et al., 2011).

The boundaries where the residual vertical velocity changes sign from positive to negative, known as the turnaround
latitudes (TL), are overlaid as thick lines in Figure 4. A quadrupling of $CO_2$ leads to a narrowing of the upwelling region
between lower and middle stratosphere (shallow branch) that maximises around ~30 hPa (Hardiman et al., 2014). This arises
predominantly from the uniform SST warming (Fig. 4c), with additional weaker contributions from the rapid adjustment and
SST patterns (Fig. 4b and 4d). In contrast, in the deep branch of the BDC the upwelling region widens particularly in the NH
(Hardiman et al., 2014). The widening of the upwelling region in the NH upper stratosphere arises almost entirely from the
rapid adjustment, while the smaller tropical widening in the SH upper stratosphere is contributed by all three components.

We move now to evaluating the changes in downwelling in the extratropics. The full $4xCO_2$ experiment shows enhanced
downwelling over the Arctic throughout the stratosphere. Both the rapid adjustment and uniform SST warming induce
comparable increases in downwelling in the Arctic, while the SST patterns do not produce significant $\overline{w}^*$ changes in this
region. In the SH, the full perturbation generates stronger downwelling in the upper stratosphere and weaker downwelling in
the middle and lower stratosphere (p>10 hPa). All three components produce increased downwelling in the Antarctic upper
stratosphere, with the largest change from the uniform SST warming and the rapid adjustment. In the lower stratosphere, the
uniform SST and SST patterns both generate reduced downwelling as seen in the full $4xCO_2$ experiment.

The relationship of the changes in residual circulation to the overall mass transport in the stratosphere can be seen in
Figure 5, which shows the annual mean residual streamfunction anomalies ($\Psi^*$) in the four experiments. The full $4xCO_2$
experiment shows enhanced poleward mass transport in both hemispheres (Hardiman et al., 2014) (Fig. 5a), with all three
components of the total forcing contributing to this pattern albeit with different magnitudes. The positive streamfunction
anomalies maximise in the NH lower stratosphere where the uniform SST warming (Fig. 5c) accounts for the majority of the
full response, with the rapid adjustment being the next largest contributor (Fig. 5b) and lastly the SST patterns generating a
small increase in poleward transport (Fig. 5d). In contrast, in the NH extratropical middle and upper stratosphere, all





components contribute similarly to the enhanced poleward mass transport. In the SH, the uniform SST warming (Fig. 5c) and to a lesser extent the SST pattern experiment (Fig. 5d), induce an acceleration of the BDC in the lower stratosphere with the latter exhibiting a response that is confined to the subtropics. Over the Antarctic polar cap in the lower and middle

stratosphere, there is no statistically significant response of the annual mean mass streamfunction in any of the perturbation experiments.

As the winter hemisphere cell of the residual circulation is the dominant one (Rosenlof, 1995), the residual mass streamfunction anomalies in the solstice seasons December-February (DJF) and June-August (JJA) are shown in the supplementary material (Figures S1 and S2, respectively). These reveal broadly similar patterns to the annual mean in both

seasons. The largest response in both hemispheres occurs in DJF, while the NH exhibits a stronger response compared to the SH. In DJF, the response to the SST patterns in the NH is confined to the subtropical lower stratosphere, while the rapid adjustment and uniform SST warming induce increase poleward transport across most of the stratosphere, with the latter showing around three times larger anomalies near the maximum in the subtropical lower stratosphere. The peak NH anomaly due to the rapid adjustment is around double that for the SST patterns in DJF. In JJA, the increase in the SH mass transport is

largely present in the subtropics while in the extratropics there is a reduction in the streamfunction which comes from both the uniform SST warming and the SST patterns. The strengthened SH poleward transport due to the uniform warming is confined to the SH subtropical lower stratosphere, while the rapid adjustment induces poleward flow that also extends into the NH that is associated with the deep branch of the BDC.

An important question is the extent to which the combined residual circulation anomalies from the rapid adjustment,

global uniform SST, and SST pattern experiments match the full 4xCO$_2$ response. This comparison is shown in Figure S3. The main differences are that the combined responses overpredict the enhanced poleward flow in the NH extratropical lower stratosphere, while there are dipole anomalies straddling the equator in the tropical mid-stratosphere associated with the differences seen in the QBO features across the experiments. Nevertheless, the differences between the linear sum of responses and the full experiment are generally small compared to the overall changes, which supports the use of the

experiments to decompose the total response into separate parts.

### 3.4 Wave forcing and downward control

To understand the changes in residual circulation shown in Figures 4 and 5, we now focus on the wave forcing in each experiment. As the distribution of wave forcing shows a strong annual cycle, we separate the changes into the winter and summer seasons in each hemisphere (DJF and JJA). Figure 6 shows the DJF average Eliassen–Palm flux divergence (EPFD)

anomalies from preindustrial for runs B, C, D and E along with the anomalous Eliassen–Palm flux vectors. The full experiment (Fig. 6a) shows increased EPF divergence in the NH extratropical upper stratosphere and in the midlatitude middle stratosphere. In the SH, there is a broad region of enhanced EPF convergence peaking around 50-60ºS over a layer spanning 3 to 70 hPa. There is a reduction in EPF convergence near the SH subtropical stratopause. Between ~50-70 hPa, there is enhanced EPF convergence in the tropics and subtropics in both hemispheres. This modulation in the location of the



maximum in the resolved wave forcing is associated with the equatorward movement of the critical layers (Fig. 3), allowing more Rossby wave activity to penetrate into the subtropical latitudes, accelerating the shallow branch of the BDC, consistent with the findings of Shepherd and McLandress (2011).

The rapid adjustment and uniform SST warming contribute similar increases in EPF convergence in the NH upper stratosphere in DJF (Figs. 6b and 6c). In the NH middle stratosphere, the uniform SST warming explains most of the

increase in EPF convergence seen in the full experiment, but the rapid adjustment does contribute in the region 20-40ºN. The uniform SST warming also contributes to most of the increase in EPF convergence in the lower to middle SH extratropical stratosphere in austral summer, but the rapid adjustment and SST pattern (Fig. 6d) do make smaller but significant contributions to the increased wave forcing in that region. The uniform SST warming produces most of the enhanced EPF convergence in the tropical and subtropical upper troposphere-lower stratosphere (UTLS).

Figure 7 shows the EPFD anomalies in JJA in each experiment. The picture in the summer NH in the full experiment is similar to that in the SH in DJF (Fig. 6a). Specifically, there is anomalous EPF divergence in the extratropical lower stratosphere and anomalous EPF convergence in the middle to upper stratosphere, representing an upward shift and extension of the region of climatological EPF convergence in this region (contours). Near the subtropical stratopause there is anomalous EPF divergence that comes mainly from the rapid adjustment (Fig. 7b). The anomalous EPF convergence in the

middle stratosphere comes mainly from the uniform SSW warming (Fig. 7b) with smaller but significant contributions from the rapid adjustment and SST patterns (Fig. 7d). In the winter SH, the picture is rather different from the NH in DJF. The full experiment shows anomalous EPF divergence in the SH upper stratosphere, which represents a weakening of the climatological EPF convergence in this region (contours). This is attributed mainly to the uniform SST warming, but there are also significant EPF convergence anomalies near the SH subtropical stratopause from both the rapid adjustment as well

as the SST patterns. The changes in EPFD in the SH middle and lower stratosphere in austral winter have a more complex structure. The full experiment shows a tripolar pattern between 30 to 70 hPa, with anomalous EPF convergence poleward of 60ºS and between 20-40ºS and anomalous divergence between 40-60ºS. This pattern is mainly reproduced in the uniform SST warming experiment but with a smaller contribution to the two regions of anomalous EPF convergence from the rapid adjustment.

The total anomalous residual circulation is driven by resolved and parameterized wave forcing. The seasonal parameterized wave forcing (NOGWD and OGWD) anomalies are shown in the supplementary material (Figures S4-S5 for DJF and Figures S6-S7 for JJA). The peak changes in parameterized wave forcing are smaller than the anomalous resolved wave forcing by around a factor of two. The anomalous NOGWD is mainly in the upper stratosphere and comes from the uniform SST warming. There is anomalous OGWD (Figs. S5 and S7) in the winter hemispheres near the edge of the polar

vortex, which has comparable contributions from the rapid adjustment and the uniform SST warming.

We now quantify the contributions of the different wave types to the anomalous mass circulation in the shallow and deep branches of the BDC. Figures 8 and 9 show latitudinal profiles of the annual mean mass streamfunction anomalies in each experiment at 70 hPa and 10 hPa, respectively, along with the DCP inferred contributions from the resolved and





parameterized components of the wave forcing. The DCP calculation for the total wave forcing underestimates the directly
calculated maximum streamfunction anomaly in the model by around 20%.

In the lower stratosphere at 70 hPa, the estimated streamfunction anomalies from the total wave forcing in the full
$4xCO_2$ experiment come mainly (>80%) from the resolved wave forcing (Fig. 8a), with a smaller and more homogeneous
contribution from the parameterized wave drag. The resolved wave forcing explains almost all of the DCP estimated
response in the rapid adjustment experiment (Fig. 8b) and most of it in the uniform SST warming (Fig. 8c) case. The
component that contributes the smallest increase in streamfunction at 70 hPa, the SST pattern experiment (Fig. 8d), shows
roughly equal contributions from parameterized and resolved wave forcing. The overall dominance of the resolved wave
forcing for the strengthened shallow branch is consistent with the larger changes in resolved wave drag in the lower and
middle stratosphere (Figures 6 and 7) compared to the parameterized wave forcing changes in this region (Figures S4-S7).

In the deep branch in the upper stratosphere, the Full $4xCO_2$ experiment shows contributions to the enhanced
streamfunction from both resolved and parameterized wave forcing (Fig. 9a). In the NH, the EPFD contribution explains
around two thirds of the total DCP estimated streamfunction anomalies and GWD around one third. The positive NH
streamfunction anomaly from EPFD poleward of 30°N comes from both the rapid adjustment (Fig. 9b) and the uniform SST
warming (Fig. 9c). In contrast, the positive streamfunction anomaly in the deep branch from parameterized wave drag comes
mainly from the uniform SST warming (Fig. 9c).

In the SH, the picture in the Full experiment is somewhat more complex, with the major contribution to the enhanced
poleward mass transport coming from GWD, which is partly offset by an opposite contribution from the EPFD. This positive
SH streamfunction anomaly associated with EPFD comes mainly from the uniform SST warming (Fig. 9c), which also
generates enhanced SH poleward transport via enhanced GWD. This increased poleward flow in the SH deep branch is
further increased by the rapid adjustment (Fig. 9b) with contributions from both resolved and parameterized wave drag. In
both hemispheres, the SST pattern has little effect on the wave forcing of the deep branch (Fig. 9d).

### 3.5 Uncertainty in global mean SST response

Figure 10 summarizes the results by showing the annual mean tropical upward mass flux anomalies in the different
experiments in the shallow and the deep branch of the BDC, at 70 hPa (Fig. 10a) and 10 hPa (Fig. 10b), respectively. Also
shown are the mass flux anomalies from the high and low uniform warming experiments (runs F and G), which span the
spread in $4xCO_2$ global mean SST response across the CMIP5 models. In the lower branch of the BDC, the annual mean
tropical upward mass flux increases by 45% in the full experiment compared to piControl ($3.1 \times 10^9$ kg s$^{-1}$). The uniform
SST warming accounts for ~70% of this increase, with the rapid adjustment (~20%) and SST patterns (~10%) contributing
the remainder in comparable amounts. The central estimates of the mass flux anomalies at 70 hPa in the three uniform SST
warming (2.1, 3.4, 4.9 K) experiments are 1.4, 2.3 and $3.4 \times 10^9$ kg s$^{-1}$, which gives an approximate linear scaling of $0.7 \times$
$10^9$ kg s$^{-1}$ K$^{-1}$ (~10% K$^{-1}$). In the shallow branch of the BDC, the uncertainty from the CMIP5 model spread in global mean
SST response to $4xCO_2$ is larger than the contribution from the rapid adjustment and the SST pattern effect.



In the deep branch at 10 hPa, the total mass flux increases by around 35% in the full experiment ($0.6 \times 10^9$ kg s$^{-1}$). This increase comes almost equally from the rapid adjustment (~40%) and the uniform SST warming (~45%), with the remaining ~15% contribution coming from the SST pattern effect. The central estimates of mass flux anomalies at 10 hPa in the three

uniform SST warming experiments are 0.17, 0.29 and $0.50 \times 10^9$ kg s$^{-1}$, which gives an approximate linear scaling with global mean SST of $0.11 \times 10^9$ kg s$^{-1}$ K$^{-1}$ (~7% K$^{-1}$). In the deep branch, the magnitude of the anomalous mass flux due to the rapid adjustment is therefore comparable to the uncertainty from the model spread in global mean SST response to 4xCO$_2$.

## 4 Discussion and conclusions

We have performed idealised experiments with the HadGEM3-A model to decompose the long-term Brewer-Dobson

circulation response to an abrupt quadrupling in CO$_2$ into three components: 1) a rapid atmospheric adjustment where CO$_2$ is added to the atmosphere but sea surface temperatures (SST) are held fixed; 2) a contribution from the global-average SST change; and 3) an SST pattern effect. The SST anomalies in response to the abrupt 4xCO$_2$ perturbation were derived from the CMIP5 multi-model ensemble. The multi-model annual mean global-mean SST anomaly over the final 50 years of the CMIP5 abrupt-4xCO$_2$ runs is 3.4 K. The SST pattern (i.e. the local deviation from the global mean value) shows relatively

warmer SST across the tropical oceans and most of the Northern hemisphere and relatively cooler SST across much of the Southern Ocean and in the northern North Atlantic. The HadGEM3-A simulations are perturbed from a reference preindustrial state, and sea ice concentrations are held fixed to enable a clean separation of the effects of SST without combining this with the potential effect of sea ice changes on the stratosphere (e.g., Kim et al., 2014).

In the tropical lower stratosphere, the 45% increase in the annual mean mass transport by the BDC under the full 4xCO$_2$

perturbation comes mainly (~70%) from the uniform SST warming (Lin et al., 2015). The remainder comes from the rapid adjustment (~20%) and the SST pattern effect (~10%). The amplitude of the SST pattern effect on the mass transport in the shallow branch is broadly comparable to that found on interannual timescales associated with ENSO (Calvo et al., 2010; Simpson et al., 2011), though note that the SST pattern imposed here is very different from a canonical ENSO SST pattern. In the upper stratosphere, where the deep branch of the BDC occurs, the increase in the BDC mass transport under abrupt-

4xCO$_2$ comes from the rapid adjustment and the uniform SST warming in roughly equal measure. The results are consistent with studies that show an important role for the strengthening of the subtropical jets under climate change (e.g., Garcia and Randel, 2008; Lin and Fu, 2013; McLandress and Shepherd, 2009; Shepherd and McLandress, 2011), which in the decomposition performed here comes mainly from the uniform SST warming. However, our results also demonstrate an important role for radiative cooling of the stratosphere by CO$_2$, which in the decomposition performed here comes mainly in

the rapid adjustment, for driving the increase in the deep branch of the BDC. This means that in transient atmosphere-ocean abrupt-4xCO$_2$ experiments, there are expected to be different characteristic timescales for the BDC response. In the lower stratosphere, the timescale of the BDC response will be mainly determined by the rate of tropospheric warming and associated changes to upper tropospheric heating and subtropical jet strength, while in the upper stratosphere there will be a





fast timescale associated with the $CO_2$ radiative cooling and a slow timescale also tied to the tropospheric warming. The
results therefore demonstrate the existence of two timescales in the response of the BDC to increasing $CO_2$, with the relative
importance of each timescale for the long-term response being height dependent.

We further examined the effect of the uncertainty in global mean SST response to increased $CO_2$, as a proxy for model
spread in equilibrium climate sensitivity. The range in the global mean SST response across the CMIP5 models is 2.1 to 4.9
K. Further experiments imposing these global uniform SST values show an increase in the shallow branch (70 hPa) upward
mass flux of $1.4 \times 10^9$ and $3.4 \times 10^9$ kg s$^{-1}$, respectively, which can be compared to the increase of $2.3 \times 10^9$ kg s$^{-1}$ in the
multi-model mean global SST experiment. In the deep branch (10 hPa), the upward mass flux change for uniform SST
warming of 2.1 and 4.9 K is $0.17 \times 10^9$ and $0.5 \times 10^9$ kg s$^{-1}$, respectively, which can be compared to $0.29 \times 10^9$ kg s$^{-1}$ in the
multi-model mean global SST experiment. Therefore, in the lower stratosphere the contribution from the uniform SST
warming and its uncertainty is larger than the rapid adjustment and SST pattern effects on the shallow branch of the BDC.
However, in the upper stratosphere the uncertainty in the magnitude of global mean SST increase across models is
comparable to the magnitude of the rapid adjustment effect on the increased deep branch of the BDC.

Using the tropical mass flux anomalies described above and the GSAT changes in the experiments given in Section 2.2,
we calculate a linear dependence of the tropical upward mass flux on GSAT of $0.62 \times 10^9$ kg s$^{-1}$ K$^{-1}$ (~9% K$^{-1}$) at 70 hPa and
$0.10 \times 10^9$ kg s$^{-1}$ K$^{-1}$ (~6% K$^{-1}$) at 10 hPa. Hardiman et al. (2014) examined CMIP5 models and calculated a multi-model
mean trend in 70 hPa upward mass flux of 3.2% decade$^{-1}$ over 2006-2099 in the Representative Concentration Pathway 8.5
(RCP8.5) emissions scenario. The multi-model mean change in GSAT between 2081-2100 relative to 1986-2005 is 3.7 K in
the RCP8.5 scenario (Collins et al., 2013). This gives an approximate multi-model mean GSAT trend for the 21$^{st}$ century of
0.35 K decade$^{-1}$. Dividing these two numbers gives an estimate for the relationship between 70 hPa mass flux and GSAT of
~9% K$^{-1}$. This is in relatively good agreement with our results, despite the different modelling approaches, though our
estimate would be slightly larger if the contributions from the rapid adjustment and SST pattern effects, which are implicitly
included in the simulations used by Hardiman et al. (2014), were accounted for. The comparison is further complicated by
the projection reduction in the BDC due to ozone recovery (e.g., Oman et al., 2009), which offsets part of the GHG-driven
increase over the 21$^{st}$ century; this effect is also included in the 21$^{st}$ century scenario simulations used by Hardiman et al.
(2014) and, if removed, this would presumably make the inferred relationship to GSAT larger than the ~9% K$^{-1}$ estimated
above based on the CMIP5 RCP8.5 scenario.

The $CO_2$ perturbation applied here is large compared to projected $CO_2$ concentrations during the 21$^{st}$ century based on
current mitigation commitments under the United Nations Framework Convention on Climate Change (UNFCCC) 2015
Paris Agreement. For a smaller increase in $CO_2$, the rapid adjustment and uniform SST warming contributions are expected
to be smaller; in this case the SST pattern effect would become proportionately more important. Our experiments have
neglected any feedbacks that induce stratospheric ozone changes; it has been shown the ozone response to 4x$CO_2$ affects the
zonal mean extratropical circulation (Chiodo and Polvani, 2017); it would be interesting to also examine the effects of ozone
on the BDC in the future. The experiments are designed to study the long-term response to an abrupt quadrupling of $CO_2$,



and they have only been performed with one model. Studies with other coupled atmosphere-ocean models that could be used to examine the transient response of the BDC to $CO_2$ would be insightful.

*Data Availability*

All model output is available from the authors upon request.

*Author Contributions*

ACM and AC designed the study. AC ran the model simulations and analysed the data. AC and ACM interpreted the data and wrote the article with input from MPC.

*Competing interests*

The authors declare no competing interests.

*Acknowledgments*

AC was supported by a University of Leeds Anniversary Postgraduate Scholarship. ACM was supported by a NERC Independent Research Fellowship grant NE/M018199/1 and the European Union's Horizon 2020 Research and Innovation
Programme under grant agreement No. 820829 (CONSTRAIN project). MPC acknowledges support through the NERC SISLAC grant NE/R001782/1. The model simulations were performed on the NERC ARCHER HPC facility. We acknowledge the World Climate Research Programme's Working Group on Coupled Modelling, which is responsible for CMIP, and we thank the climate modelling groups (listed in Table S1 of this paper) for producing and making available their model output. The analysis and visualization of the study were performed using the NCAR Command Language (NCL).

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

| Run | ID | $CO_2$ | SSTs (prescribed) |
|---|---|---|---|
| A | piControl | Pre-industrial | Pre-industrial |
| B | Full 4x$CO_2$ | 4x$CO_2$ | 4x$CO_2$ (CMIP5) |
| C | Atmos | 4x$CO_2$ | Pre-industrial |
| D | SST UW | Pre-industrial | 4x$CO_2$ (UW) – 3.4 K |
| E | SST patterns | Pre-industrial | 4x$CO_2$ (patterns) |
| F | SST UW low | Pre-industrial | Low 4x$CO_2$ – 2.1 K |
| G | SST UW high | Pre-industrial | High 4x$CO_2$ – 4.9 K |

**Table 1: The sensitivity experiments used in this study with the atmospheric $CO_2$ and the SSTs used as boundary conditions. All other boundary conditions are as in piControl.**





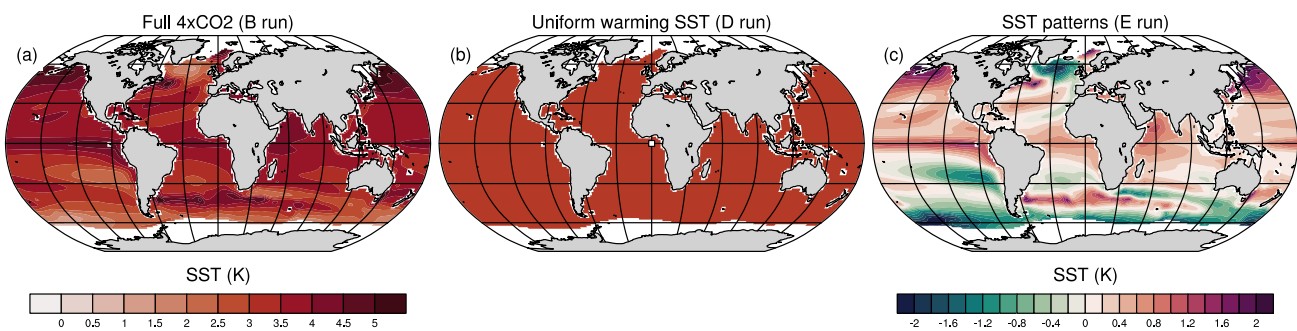

**Figure 1: Prescribed annual mean SST anomalies [K] with respect to the piControl climatology in the (a) full 4xCO₂, (b) uniform SST warming and (c) SST pattern perturbation experiments.**


Annual zonal mean T anomalies

Figure 2: Latitude vs. pressure cross-sections of annual and zonal mean temperature anomalies [K] with respect to the piControl simulation for the (a) 4xCO₂, (b) rapid adjustment, (c) Uniform SST warming, and d) SST pattern experiments. Contours show the piControl climatology. Stippling denotes where the differences are not statistically significant at the 95% confidence level. Thick yellow and black lines indicate the tropopause pressure levels in each perturbation run and in the reference simulation, respectively.


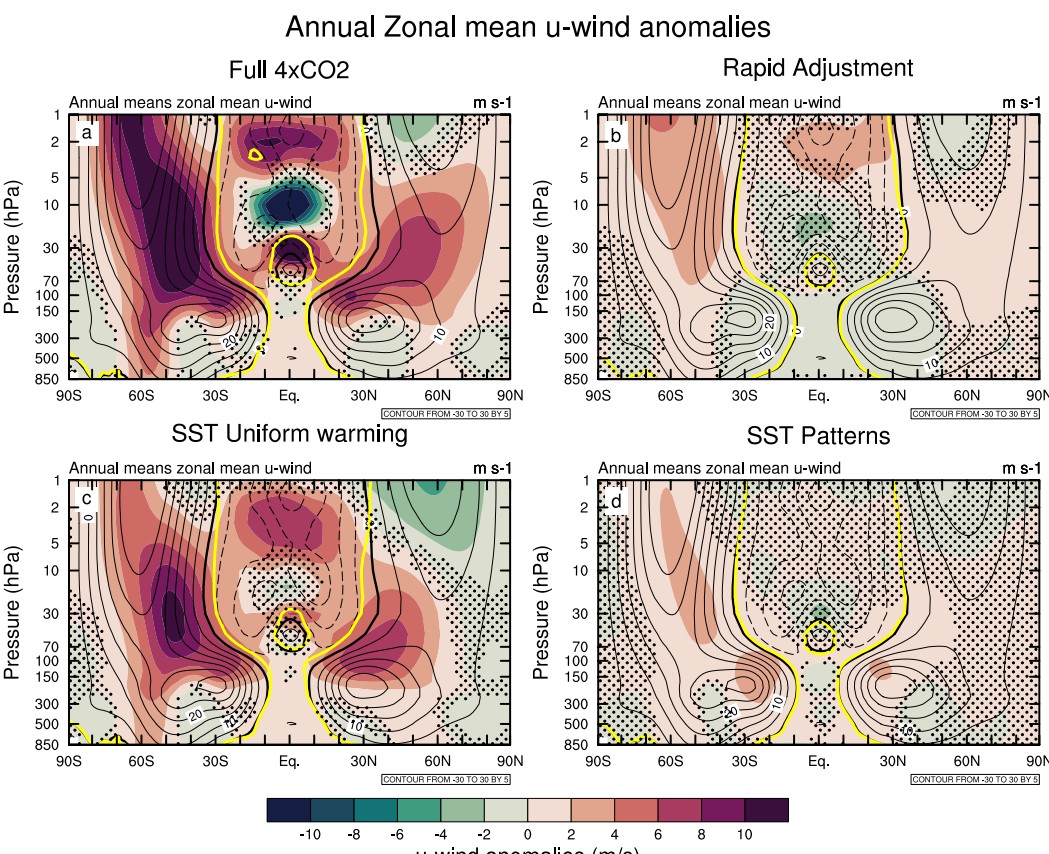

**Figure 3: As in Figure 2, but for the annual and zonal mean zonal wind anomalies [m s⁻¹]. Contours show the piControl climatology. The thick black lines denote the critical lines (ū = 0) in piControl and the thick yellow lines for each perturbation experiment, respectively.**




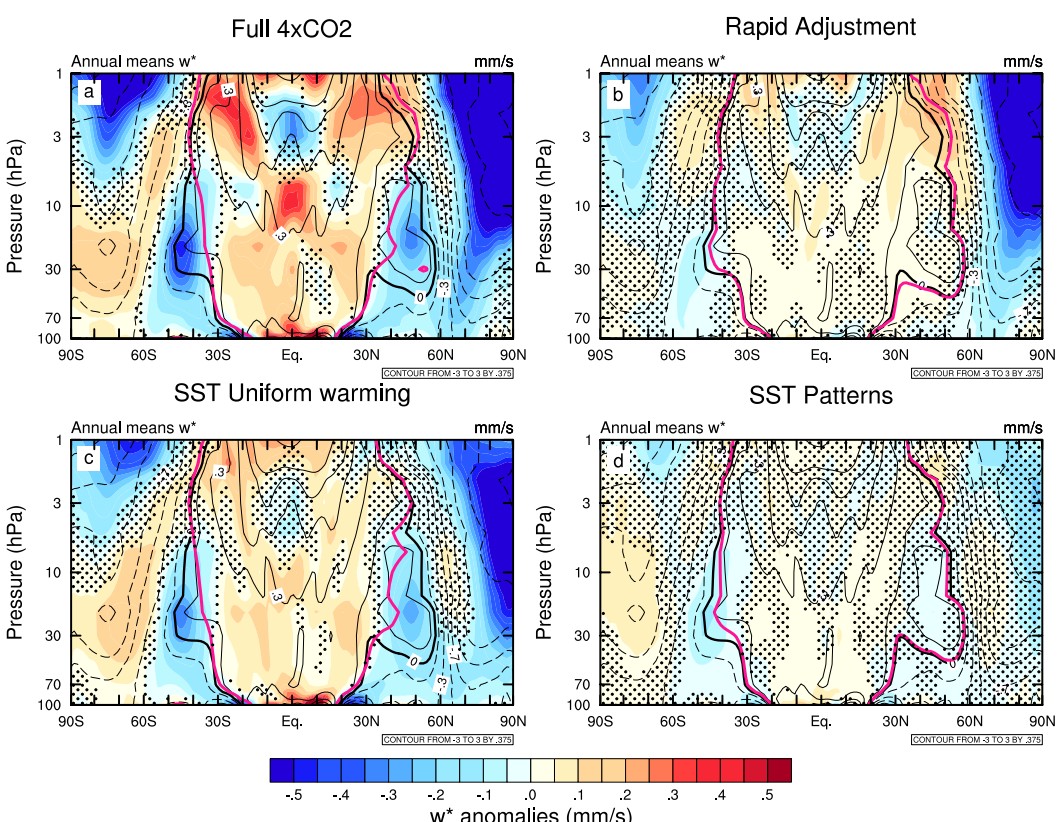

**Figure 4: As in Figure 2, but for the annual mean TEM residual vertical velocity anomalies [mm s⁻¹]. Contours show the piControl climatology and range from -3 to 3 mm s⁻¹ in increments of 0.375 mm s⁻¹. The thick black lines denote the turnaround latitudes ($\overline{w}^{*}$ = 0) in piControl and pink thick lines for each perturbation experiment, respectively.**




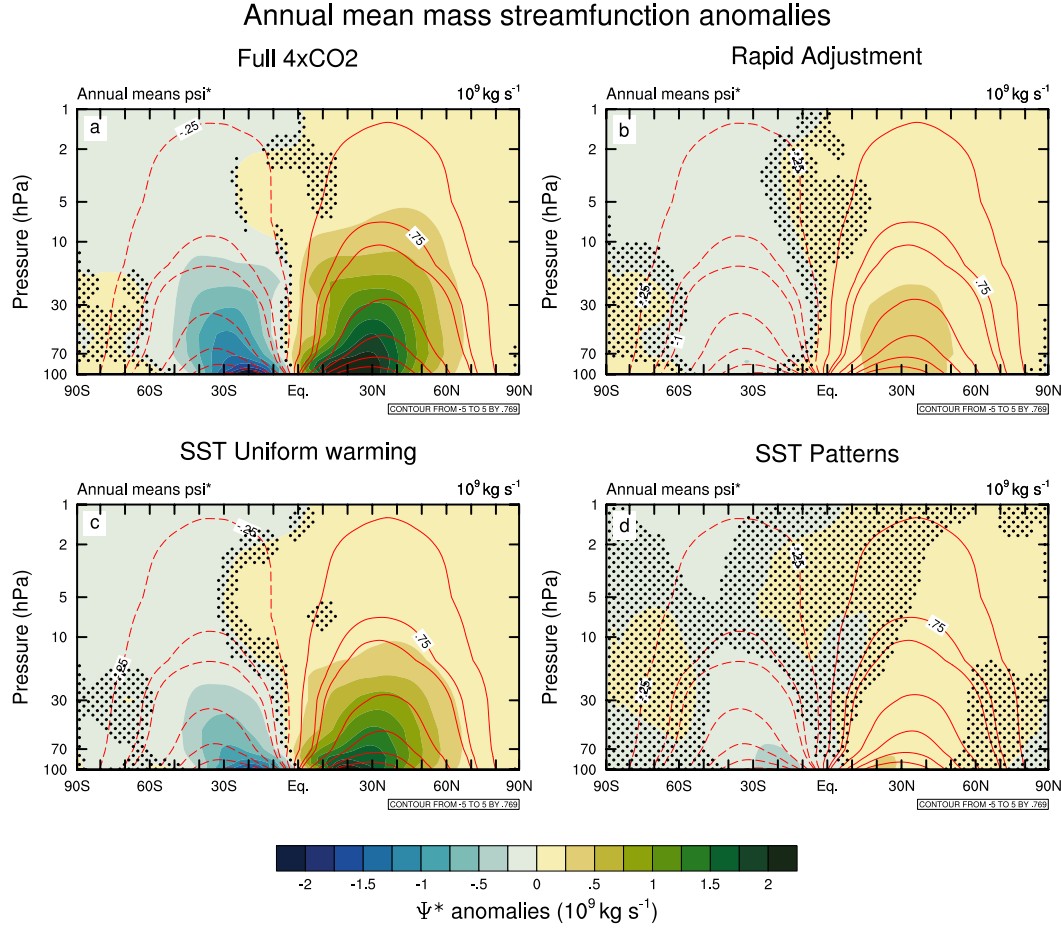

**Figure 5: Annual mean residual mass streamfunction anomalies [109 kg s-1] in the decomposed 4xCO2 perturbation experiments. Stippling denotes where the differences are not statistically significant at the 95% confidence level. Red contours plotted at -5, -4, -3, -2, -1, -0.75, -0.25, 0.25, 0.75, 1, 2, 3, 4 and 5 × 109 kg s-1 show the piControl climatology with negative values showed in dashed contours.**




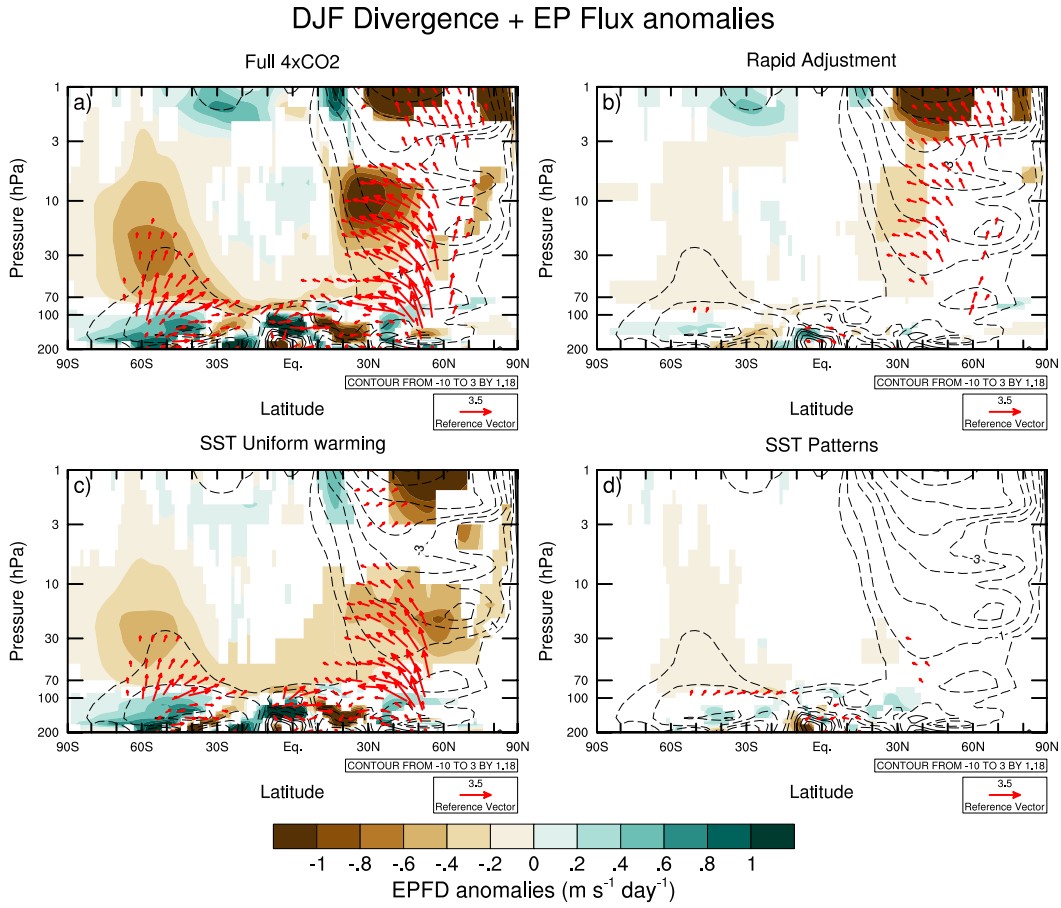

**Figure 6: DJF average EP flux vector (red arrows) [m² s⁻²] and EP flux divergence anomalies [m s⁻¹ day⁻¹] (shading) in the decomposed 4xCO₂ experiments. Contours show the piControl climatology with contours plotted at -10, -8, -6, -4, -3, -2, -1, -0.5, 0.5, 1, 2, 3 m s⁻¹ day⁻¹. The EP flux vector and divergence anomalies are only plotted where they are significant at the 95% confidence level. The EP flux vectors have been scaled following Edmon et al. (1980).**



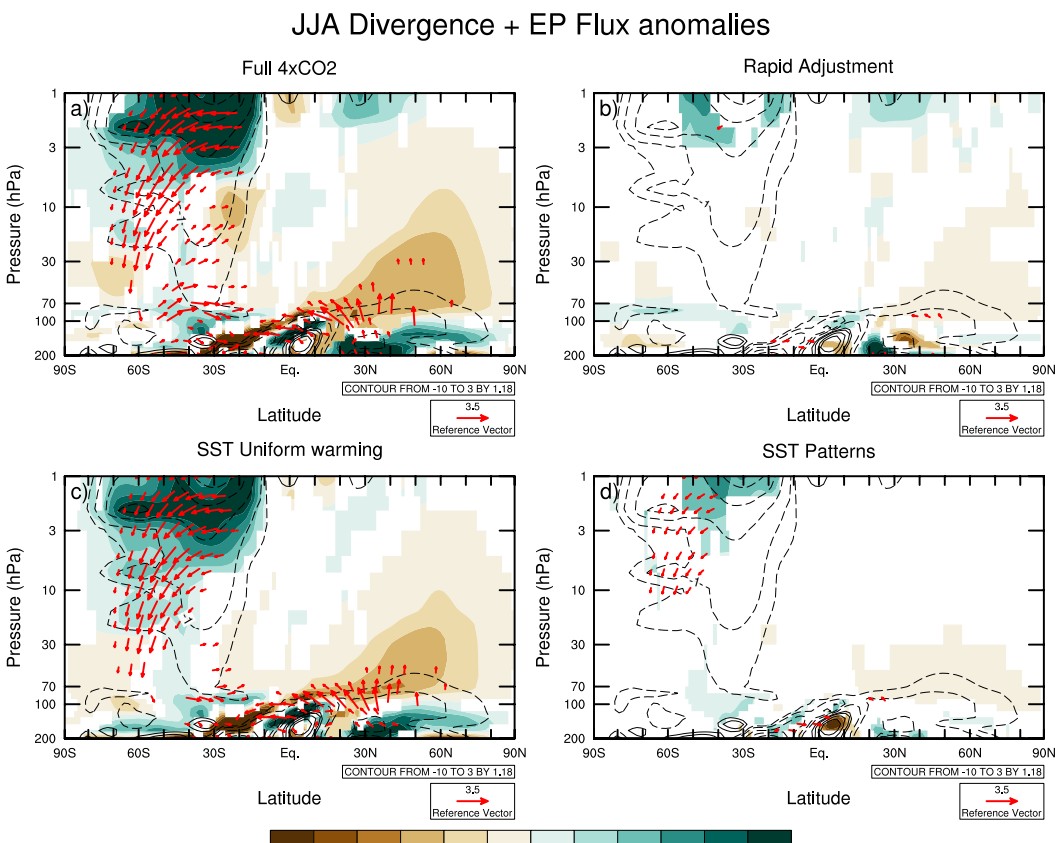

Figure 7: As in Figure 6, but for the JJA season.



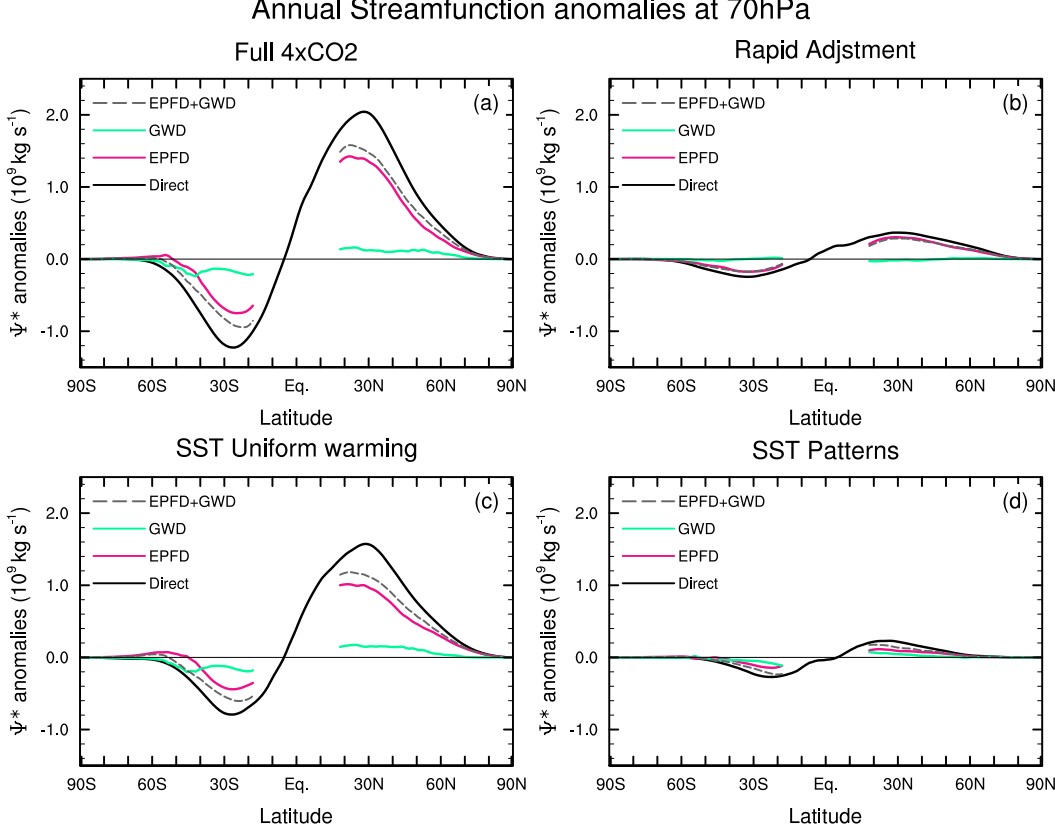

Figure 8: Annual mean residual streamfunction anomalies [$10^9$ kg s$^{-1}$] at 70 hPa in the decomposed 4xCO₂ perturbation experiments. Black line shows the direct calculation, the downward control calculations for EPFD, OGWD + NOGWD and their sum (EPFD + OGWD + NOGWD) are shown in magenta, green and grey dashed, respectively.




Figure 9: As in Figure 8, but at 10 hPa.






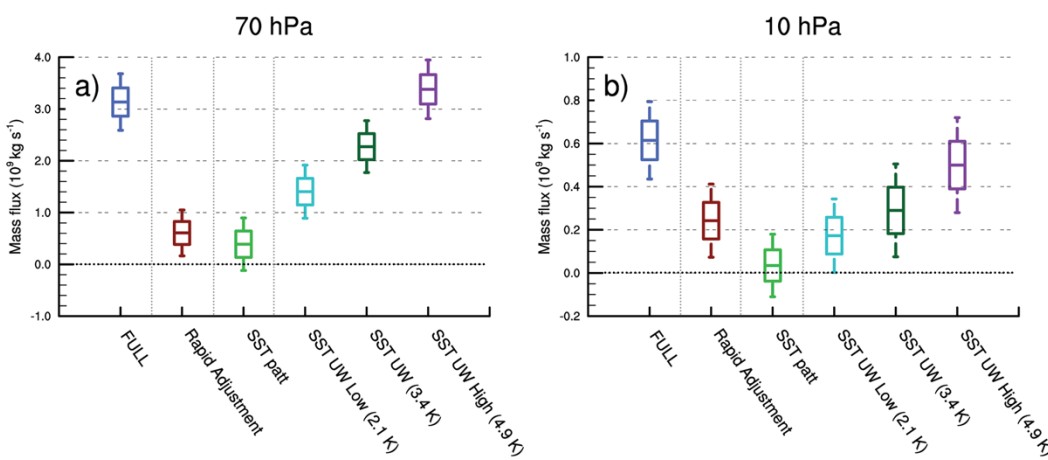

**Figure 10:** Annual mean tropical upward mass flux anomalies [$10^9$ kg s$^{-1}$] at (a) 70 hPa and (b) 10 hPa in the different perturbation
experiments as labelled. The edges of the boxplots indicate ±1 standard deviation of the interannual variability and the whiskers
indicate ±2 standard deviations.