# Peer review of "Decomposing the response of the stratospheric Brewer-Dobson circulation to an abrupt quadrupling in CO2"

_Weather and Climate Dynamics, 2020_

## Referee Comment (RC1) · Anonymous Referee #1 · 14 Feb 2020

General comments:

This is an interesting, relevant and well-presented paper. The authors consider the drivers behind the response of the BDC to 4x CO2 using 50-year simulations of HadGEM3. As such, the paper serves as an experimental report. The authors decompose the components of the response into those arising from rapid adjustment (holding sea-ice and SSTs constant at pre-industrial values), the global-mean SST warming (relative to pi-control), and the specific pattern of SST warming (global mean removed). I can recommend the paper for publication with a few changes as I have outlined below, none of which are particularly major and most of which just improve the

readability and flow of the manuscript (to allow the science to stand out). I have a few specific comments which pertain more to the choices the authors have made in what they show/do.

Specific comments:

L47: I think "turnaround latitudes" needs to be briefly explained here.

L57: I am not sure why the more general "tropical waves" is mentioned and then elaborated as "equatorially trapped quasi-stationary Rossby waves". One is more general, while the other is more specific. Only one is necessary.

L140: Would it not have also been possible to perform an experiment where sea ice is allowed to vary? Comparing the results of this perturbation with those where it is held constant could show some of the possible effects on the BDC to which the authors allude is "uncertain".

L242-245: This mention of changes to the QBO is tantalising! Would it be possible to include this in at least supplementary figures? This is up to the authors, and I agree it is not the focus of the study, but this mentioning of it without further information leaves me wondering what stones are unturned.

L403: The statement "the SST pattern imposed here is very different from a canonical ENSO SST pattern" confused me. Is it? Does this refer to the global pattern, or the tropical Pacific in particular? This seems important given what is mentioned on L90.

L384/Figure 10: the SST pattern effect result is non-significant (the confidence interval overlaps with 0). This should probably be mentioned.

Method: What method was used to determine the 95% confidence levels? I don't think the authors have stated this.

Results: There are cases where the individual component results are compared in a qualitative sense (e.g. L280), but it would be useful if these were sometimes more

quantitative (e.g. X% more. . .) like in section 3.5.

Figures: In general I did not notice that the bottom y-limit changes quite a bit between each figure, as they are all in the same format. Perhaps worth mentioning in the captions.

Technical corrections:

L75: GEOS is not defined here. It is not particularly important, but it stands out as all other acronyms are.

L90: Tilde is missing from Niño.

L90: Although it would be common to say "ENSO itself" and not "The ENSO. . .", so I understand why the authors have written it in this way, I think this sentence should begin with "THE El Niño Southern Oscillation. . .".

L91: Capital H on Hemisphere (also elsewhere)

L95: Definite article is missing ". . .using THE Whole Atmosphere. . ."

L109: In the list of the components, the phrasing on (2) is slightly different to the other two, and different to how it is in the abstract which makes it 'flow' less well. Consider using the phrasing as used in the abstract "a contribution from. . ."

L172: Missing capital I on McIntyre (also in the references)

L173: The final Andrews citation should probably be non-parenthetical ". . .defined following Andrews et al. (1987):. . ."

L175/equation 1: are the dots necessary? These are not consistently used in the equations shown in this paper and are not standard for scalar multiplication.

Equation 4: The integral is missing the variable of integration (dz')

L204: "maximum" should be "maximised", and a mention of by how much would maybe be good here L209: Which figure panel is run C?

L211-213: Is this sentence describing how the greenhouse effect works really needed?

L230: Here, and elsewhere, additional hyphenation increases readability. For example, please revise to "annual-mean zonal-mean zonal wind".

L256: Insert "but small", for "significant, but small, increases..."

L263 & 266 & 267: Why is Hardiman et al. cited for results that are in the figures? If it is to say that the result is consistent, then please state as such.

L272: Consider changing p<10 hPa to "below 10 hPa" for readability.

L310: Eliassen-Palm Flux is earlier abbreviated to EPF.

L405-410: "important role" and "decomposition performed here" and both repeated twice in the same paragraph. Consider revising one of each to a different phrase.

L437: "the projection reduction" should be "the projected reduction"

L634: This reference has two hyperlinks.

L637: Is this cited in the text? NCL is credited in the acknowledgments but is not linked to this reference.

Figure 1: For (b), some specification that the contour value is 3.4 K would be helpful.

Figure 2 (and similar): It would be helpful if the figures had the experiment labels on them, or in the caption, as this can get confusing.

Figure 6: This shows EP flux vector ANOMALIES which is not stated in the caption.

---

## Referee Comment (RC2) · Anonymous Referee #2 · 14 Feb 2020

This study is on the changes in the stratospheric Brewer-Dobson circulation in response to a quadrupling of CO2 concentration. Utilizing the HadGEM3-A model, the authors separate out the fast response to CO2 increase from the effect of uniform SST warming as well as the SST warming pattern. It is found that the uniform warming dominates the circulation changes in the lower stratosphere, but the rapid adjustment makes comparable contribution to the uniform warming at the 10 hPa. This is a useful study in understanding the climate changes in the stratosphere. The manuscript is generally well written and logically organized. I have some relatively minor comments on some of the results and discussion, and would recommend publication after the authors address these comments.

Main comments:

1. The mechanism for the rapid adjustment of the BDC. Can the authors elaborate a little more on how the rapid adjustment affect the BDC? The authors briefly mentioned the radiative cooling in the stratosphere. One possibility is that the radiative cooling then affect the strength of the polar night jet via thermal wind balance, which then affect wave dissipation and the BDC. However, the rapid adjustment of the zonal wind shows weak decrease in the NH upper stratosphere and moderate increase in the SH upper stratosphere (Fig. 3b). Such wind changes seem to be inconsistent with the changes in w* (Fig. 4b), which shows large strengthening of the downwelling over the Arctic and small strengthening over the Antarctic.

2. The EP flux divergence plots seem to be inconsistent with the changes in stream function. The downward control principle indicates that the latitudinal distribution of Psi* anomalies should be roughly consistent with those of EP flux divergence anomalies. However, the EP flux divergence anomalies seems to locate much poleward than Psi* anomalies, especially in DJF over the Southern Hemisphere (Fig. 6 vs. Fig. S1, Fig. 7 vs. Fig. S2). Based on the argument that stronger subtropical jets following warming allow more waves penetrate into the stratosphere, one would expect the anomalous wave dissipation to occur at the subtropics. This also seems to disagree with the pattern shown in Fig. 6 and 7, where maximum wave dissipation occurs around 50-60 degrees.

3. Model bias in climatology. The model simulated turn-around latitude in piControl climatology seems to be too poleward compared to reanalysis or other models (e.g., Fig. 1 in Hardiman et al. (2014)). At around 20 hPa, there are multiple turn-around latitude in the Nothern Hemisphere, indicating more than one cell of the circulation, which seems unreasonable. Such bias in the mean circulation structure reflects bias in the waves forcing and/or other background condition. Would such model bias affect the model's ability to simulate the circulation changes in response to CO2 increase?

Other minor comments and typos:

Line 152-155: experiment "A, C, B, D" should be "B, D, C, E".

Line 281-282: It is hard to compare the three components over NH extratropical middle and upper stratosphere in Fig. 5, as they all seem to be less than the contour interval. On the other hand, the w* changes shown in Fig. 4 seem to suggest that the SST pattern effect is much weaker than the uniform warming or the rapid adjustment.

Line 330: Fig. 7b should be Fig. 7c.

---

## Referee Comment (RC3) · Anonymous Referee #3 · 27 Feb 2020

Changes in the Brewer-Dobson circulation (BDC) due to increased CO2 levels are studied by distinguishing the response to CO2 changes in the atmosphere only, globally uniform changes in SSTs, and SST pattern changes. The former corresponds to the rapid-adjustment of the climate system when CO2 levels are increased abruptly. The latter two correspond to long-term changes due to the longer time scales of the oceanic response. These effects are studied consistently by using a single state-of-the-art climate model (HadGEM3-A). The BDC generally increases in strength due to increased CO2. The authors find that in the lower stratosphere the majority of this BDC strengthening can be attributed to globally uniform SST increase. In the upper stratosphere the changes due to rapid adjustment are of similar magnitude. The authors

furthermore estimate a linear sensitivity of the change in BDC strength as a function of global surface warming of roughly 9 %/K in the lower stratosphere and 6 %/K in the upper stratosphere.

Overall, the paper is well-written and the results are straightforward. I have a few general comments that I hope will help the authors to sharpen their discussion and to better put the work into broader context. Other than that I only have minor comments; once these comments have been taken into account this manuscript should be publishable.

General comments:

SST pattern changes and ENSO: there are frequent remarks about the resulting BDC changes from the SST pattern changes to be similar to ENSO-induced anomalies. However, in the discussion section (line 403) the authors remark that "the SST pattern imposed here is very different from a canonical ENSO SST pattern". If that is the case, isn't it surprising then that the BDC changes due to the SST pattern changes look similar to those due to ENSO? To me this calls for corresponding discussion/elaborations somewhere in the manuscript.

Shallow versus deep branch changes: it seems that the authors interpret changes in upwelling strength through 70 hPa as representative of the shallow BDC branch, whereas those at 10 hPa as representative of the deep branch. Although it is certainly true that there isn't a clear vertical level where the shallow branch stops and the deep branch takes over, perhaps a useful distinguishing factor is related to the tropical pipe concept (much less meridional exchange at pressure levels within the tropical pipe than below). The shallow branch could then be interpreted as being primarily confined to that part of the BDC that involves strong meridional dispersion below the bottom of the tropical pipe. My recollection is that this level (bottom of tropical pipe) is close to 70 hPa, although this may vary from model to model. By that argument the upwelling through 70 hPa is more a measure of the deep rather than the shallow BDC branch and neither of the quoted upwelling changes correspond to the shallow branch strength. I

admit that all of this may be a bit philosophical, but the authors may wish to include a bit of discussion on this point. This isn't an issue when simply referring to 70 vs. 10 hPa without the connotation of shallow vs. deep branch. But even in that case, one wonders about BDC changes in the lower half of the stratosphere (assuming a global mean tropopause pressure somewhere around 150 hPa, roughly half of the stratosphere is located below 70 hPa) ...

Seasonal versus annual means: residual circulation changes are shown in terms of annual means in the main manuscript, whereas those of EP flux divergence are shown in terms of seasonal means. Line 308 presents a specific argument in favour of seasonal means. I didn't understand why this argument should apply to the wave forcing but not the resulting residual circulation, hence why seasonal means were delegated to the supplement in the case of the residual circulation? Please clarify somewhere.

Minor comments:

line 47: not sure I can follow the argument here - why couldn't the wave forcing around the turnaround latitudes change if there was a change in wave activity from the troposphere?

line 51: the Randel and Held reference is appropriate for the connection of wind pattern and critical levels, but I don't think these authors talked about an upward movement of critical levels due to climate change; so the placement of this reference may be misleading

line 81: GEOS-CCM model: the acronym "CCM" already contains "model"

line 88: here and at other place: "warmer SSTs" should be something like "higher SSTs" (or "warmer sea surfaces")

line 106: "three distinct effects" – would be good to briefly remind reader about the effects

line 144: is the runtime for the 4xCO2 experiments long enough to call the final state

"quasi-equilibrium"? I could imagine that there's still drift due to ocean response, even after 150 years.

line 147/148: please add comment about the 3.4 K warming, especially comparing it to the equilibrium response to 4xCO2 (which should be double the climate sensitivity if I understand correctly, so the 3.4 K value seems small)

line 175 and following: please clarify use of vertical coordinate; your model runs in height coordinates (not log-p height), but the TEM diagnostics are formulated in log-p height – was this done by first interpolating the data?

line 181: I assume that Eq. 2 is only integrated to the respective vertical level of interest (so that Psi* is still function of log-p height), not all the way to the surface (unless for lowest level)? Also, your definitions in Eqs. 1 and 2 are circular: Eq. 1 requires knowledge of Psi* and Eq. 2 requires knowledge of v* ... even though these are standard diagnostics, it would be more helpful to define the residual streamfunction first based on the vertically integrated v and the heat flux contribution; then define v* and w* (or, alternatively, define v* and w* in terms of v and w + heat flux contribution; then use Eq. 2 for Psi*).

Eq. 4: the integral is missing a "dz"

line 244: please make more definite statements about the direction of changes of these QBO characteristics (or omit the comment altogether)

line 259: this is a good example where you make a reference to ENSO-like SST perturbations, but fall short in discussing how your SST pattern changes actually do correspond to ENSO (or not)

line 263: this is a good example where I found your reference to the shallow BDC branch confusing – to me it doesn't really extend to 30 hPa

lines 299/300: should this result perhaps be shown / referred to right away with the methods section? Also: it sounds a bit misleading to me to start the paragraph with

"An important question" and then talk about results shown in the supplement – if they really are important, why aren't they shown in the main part of the paper?

line 330: SSW -> SST

line 350: could you elaborate where this 20 % disagreement could come from?

line 400: the statement is based on results from this paper, so I assume the reference to Lin et al. is meant to state that they found similar results? Please clarify

line 434: this value ($\sim$9 %/K) is exactly equal to the one you quote for your results, so the agreement is exact (or almost exact) and not just "relatively good" - am I missing something?

line 675 (Fig. 3 caption): the u=0 lines are only critical lines for stationary waves - please clarify

Fig. 4 and related discussions: visually, it doesn't look like the positive anomalies compensate the negative anomalies on a given pressure level (perhaps when scaled by surface area they do), but shouldn't they based on mass conservation? Did you check?

---

## Author Comment (AC1) · 30 Mar 2020

**Author response to referee comments on Chrysanthou et al. (2020) "Decomposing the response of the stratospheric Brewer-Dobson circulation to an abrupt quadrupling in CO$_2$" submitted to Weather and Climate Dynamics Discussions**

**Reply to Anonymous Referee #1**

**General comments:**

This is an interesting, relevant and well-presented paper. The authors consider the drivers behind the response of the BDC to 4xCO2 using 50-year simulations of HadGEM3. As such, the paper serves as an experimental report. The authors decompose the components of the response into those arising from rapid adjustment (holding sea-ice and SSTs constant at pre-industrial values), the global-mean SST warming (relative to pi-control), and the specific pattern of SST warming (global mean removed). I can recommend the paper for publication with a few changes as I have outlined below, none of which are particularly major and most of which just improve the readability and flow of the manuscript (to allow the science to stand out). I have a few specific comments which pertain more to the choices the authors have made in what they show/do.

Thank you for your positive comments and suggestions for improving the readability of the study. We reply to the specific points raised below in red.

**Specific comments:**

L47: I think "turnaround latitudes" needs to be briefly explained here.

Thanks for this suggestion. The first mention of the turnaround latitudes is on line 44 - we have updated the text there.

L57: I am not sure why the more general "tropical waves" is mentioned and then elaborated as "equatorially trapped quasi-stationary Rossby waves". One is more general, while the other is more specific. Only one is necessary.

Thanks for catching this. We have kept the more general reference to tropical waves.

L140: Would it not have also been possible to perform an experiment where sea ice is allowed to vary? Comparing the results of this perturbation with those where it is held constant could show some of the possible effects on the BDC to which the authors allude is "uncertain".

As suggested, we have performed a further simulation in which sea ice + SSTs are changed based on the multi-model mean CMIP5 4xCO2 experiment. We compare this "full 4xCO$_2$ + sea-ice loss" with the full 4xCO$_2$ experiment (run B) as described in the manuscript. The annual mean w* anomalies due to the changes in sea ice are shown below in Figure R1. This shows no significant changes in w* due to the additional effect of sea ice loss. We have added a comment in the Methods to say we have tested the sea ice effect and it was negligible.

[Figure]

Figure R1: Annual mean w* anomalies (mm/s) due to 4xCO₂ sea ice changes. Anomalies are calculated as the difference between the "full 4xCO₂ + sea ice loss" and "full 4xCO₂" (run B) experiments. Stippling denotes differences that are not significant at the 95% confidence level.

L242-245: This mention of changes to the QBO is tantalising! Would it be possible to include this in at least supplementary figures? This is up to the authors, and I agree it is not the focus of the study, but this mentioning of it without further information leaves me wondering what stones are unturned.

We agree this point is also interesting, but we feel it would be a separate study in its own right and that to offer enough detail to satisfactorily explain the QBO changes would detract from the main focus of the paper. We have therefore removed the mention of the changes in QBO and defer this for future study.

L403: The statement "the SST pattern imposed here is very different from a canonical ENSO SST pattern" confused me. Is it? Does this refer to the global pattern, or the tropical Pacific in particular? This seems important given what is mentioned on L90.

Thanks for pointing this out. The issue of the SST pattern was also raised by reviewer 3. On reflection the description was an oversimplification. ENSO SST anomalies are confined to the tropical Pacific, whereas the 4xCO₂ SST pattern shows features globally (by construction). This includes relatively higher SST across the tropical oceans and North Pacific and relatively cooler SSTs in the Southern Ocean. We have amended the text in the Methods to provide a more nuanced discussion about the features of the 4xCO₂ SST pattern (lines 163-165). We still discuss the BDC response to ENSO but only as a point of comparison in terms of the magnitude of its effect (tracked changes manuscript lines 461-462).

L384/Figure 10: the SST pattern effect result is non-significant (the confidence interval overlaps with 0). This should probably be mentioned.

Thank you for pointing this out. We have added this point to the text: "though the latter is not statistically distinguishable from internal variability".

Method: What method was used to determine the 95% confidence levels? I don't think the authors have stated this.

Thank you for spotting this. The statistical significance was computed based on a two-tailed Student's t-test. We have now updated all relevant figure captions (Figures 2, 5 and 7) to include the statistical method of calculating the 95% confidence.

Results: There are cases where the individual component results are compared in a qualitative sense (e.g. L280), but it would be useful if these were sometimes more quantitative (e.g. X% more: : :) like in section 3.5.

This surely improves the comparison and brings out a more quantitative sense of the responses of all perturbations. We have now quantified the contributions of each component in the annual-mean residual circulation response (Figure 4) as well as in the seasonal mean mass streamfunction anomalies (Figures 5 and 6) in this part of the text to reflect your suggestion.

Figures: In general I did not notice that the bottom y-limit changes quite a bit between each figure, as they are all in the same format. Perhaps worth mentioning in the captions.

Thank you for pointing this out, we have now updated the relevant figure captions to note the pressure range in the y-axis.

**Technical corrections:**

L75: GEOS is not defined here. It is not particularly important, but it stands out as all other acronyms are.

We have added the definition for GEOS.

L90: Tilde is missing from Niño.

Thanks for this, we have now corrected it.

L90: Although it would be common to say "ENSO itself" and not "The ENSO: : :", so I understand why the authors have written it in this way, I think this sentence should begin with "THE El Niño Southern Oscillation: : :".

Thanks for this, it makes more sense to add a "The" in the beginning of the sentence.

L91: Capital H on Hemisphere (also elsewhere)

Thanks for this, we have now changed this here and in line 95.

L95: Definite article is missing ": : :using THE Whole Atmosphere: : :"

Added "the" before mentioning WACCM.

L109: In the list of the components, the phrasing on (2) is slightly different to the other two, and different to how it is in the abstract which makes it 'flow' less well. Consider using the phrasing as used in the abstract "a contribution from: : :"

Thanks for this suggestion, we have now updated the text to reflect the listing of the components as seen in the abstract, as per your recommendation.

L172: Missing capital I on McIntyre (also in the references)

Thank you for spotting this, we have now corrected this.

L173: The final Andrews citation should probably be non-parenthetical ": : :defined following Andrews et al. (1987):: : :"

Again, thanks for spotting this, we have corrected this.

L175/equation 1: are the dots necessary? These are not consistently used in the equations shown in this paper and are not standard for scalar multiplication.

We agree that this was not consistent with the rest of the mathematical formulations of the study, so we removed the dots.

Equation 4: The integral is missing the variable of integration (dz')

Thanks for spotting this, we have now added it.

L204: "maximum" should be "maximised", and a mention of by how much would maybe be good here

We corrected this to "maximised" and added the amount of tropospheric warming induced in that vicinity (~ 8 K).

L209: Which figure panel is run C?

This was an oversight which was corrected. Thanks!

L211-213: Is this sentence describing how the greenhouse effect works really needed?

We agree that it is redundant, so we removed it.

L230: Here, and elsewhere, additional hyphenation increases readability. For example, please revise to "annual-mean zonal-mean zonal wind".

Thanks for this suggestion, we have corrected this here and elsewhere in the text.

L256: Insert "but small", for "significant, but small, increases: : :"

Thank you for this suggestion, we have implemented it in the text.

L263 & 266 & 267: Why is Hardiman et al. cited for results that are in the figures? If it is to say that the result is consistent, then please state as such.

We have removed these citations.

L272: Consider changing $p < 10$ hPa to "below 10 hPa" for readability.

Thanks for this suggestion, we have updated the text.

L310: Eliassen-Palm Flux is earlier abbreviated to EPF.

We have now corrected this, thanks.

L405-410: "important role" and "decomposition performed here" and both repeated twice in the same paragraph. Consider revising one of each to a different phrase.

We have altered the second sentence to reflect your suggestion, which now reads: "*However, our results demonstrate that an increase of the BDC in the upper stratosphere comes mainly from the radiative cooling of the stratosphere by CO2, as seen in the rapid adjustment component of the response*"

L437: "the projection reduction" should be "the projected reduction"

Nice catch, thanks.

L634: This reference has two hyperlinks.

We removed the additional hyperlink, thanks.

L637: Is this cited in the text? NCL is credited in the acknowledgments but is not linked to this reference.

It was only cited in the acknowledgments however the bibliography entry was wrong, thanks for spotting it. We have now corrected this.

Figure 1: For (b), some specification that the contour value is 3.4 K would be helpful.

Thank you for this, we have now added a string on Fig. 1b that states the value of the constant contour value of 3.4 K. The colour of the contour reflects the colormap of Fig. 1a.

Figure 2 (and similar): It would be helpful if the figures had the experiment labels on them, or in the caption, as this can get confusing.

Thank you for this suggestion. We have updated with the experiment labels the captions of all related figures to clarify which panel corresponds to which experiment.

Figure 6: This shows EP flux vector ANOMALIES which is not stated in the caption.

We appreciate that you spotted this, we now state that these are the EPF vector anomalies in the newly added DJF-mean Figure 7 caption.

---

## Author Comment (AC2) · 30 Mar 2020

Author response to referee comments on Chrysanthou et al. (2020) "Decomposing the response of the stratospheric Brewer-Dobson circulation to an abrupt quadrupling in CO2" submitted to Weather and Climate Dynamics Discussions

**Reply to Anonymous Referee #2**

This study is on the changes in the stratospheric Brewer-Dobson circulation in response to a quadrupling of CO2 concentration. Utilizing the HadGEM3-A model, the authors separate out the fast response to CO2 increase from the effect of uniform SST warming as well as the SST warming pattern. It is found that the uniform warming dominates the circulation changes in the lower stratosphere, but the rapid adjustment makes comparable contribution to the uniform warming at the 10 hPa. This is a useful study in understanding the climate changes in the stratosphere. The manuscript is generally well written and logically organized. I have some relatively minor comments on some of the results and discussion and would recommend publication after the authors address these comments.

We appreciate your constructive comments in order to improve the results and discussion of the study. We reply to the specific points raised below in red.

**Main comments:**

1. The mechanism for the rapid adjustment of the BDC. Can the authors elaborate a little more on how the rapid adjustment affect the BDC? The authors briefly mentioned the radiative cooling in the stratosphere. One possibility is that the radiative cooling then affects the strength of the polar night jet via thermal wind balance, which then affect wave dissipation and the BDC. However, the rapid adjustment of the zonal wind shows weak decrease in the NH upper stratosphere and moderate increase in the SH upper stratosphere (Fig. 3b). Such wind changes seem to be inconsistent with the changes in w\* (Fig. 4b), which shows large strengthening of the downwelling over the Arctic and small strengthening over the Antarctic.

Thanks for this interesting comment. The thermal wind response due to the direct cooling is fairly small because the CO2 radiative cooling is quite homogeneous in latitude (Fels et al., 1980). One hypothesis that we think is plausible is that the radiative cooling alters the refractive index in the stratosphere, which changes the propagation and breaking of Rossby waves. The refractive index, nr, is dependent on the Brunt-Väisälä frequency (e.g., Matsuno, 1970), N2, which is in turn related to the vertical gradient of potential temperature:

$$N^2 = \frac{g}{\theta} \frac{d\theta}{dz}$$

Hence the stratospheric cooling, which increases with height, alters N2 and in turn nr. Unfortunately, we do not have the model variables to calculate nr directly for the simulations. We have added a section discussing this possible mechanism (revised manuscript lines 350 - 359 / tracked changes manuscript lines 383-391):

"Previous studies have demonstrated mechanisms for tropospheric warming to influence the stratospheric EPFD and residual circulation (e.g., Shepherd and McLandress, 2011), but the mechanism through which the rapid adjustment acts on EPFD in the upper stratosphere is less well understood. The radiative cooling in the stratosphere due to increased CO2 is relatively uniform in latitude (Fels et al., 1980), so we do not expect large direct changes in zonal wind through thermal wind balance. However, the temperature response to CO2 represents a weakening of the vertical temperature gradient, particularly in the upper stratosphere where the cooling is larger. The characteristics for wave propagation and refraction can be

quantified using a measure of refractive index (e.g., Matsuno, 1970) that is dependent on the Brunt-Väisälä frequency ( $N_2 = g/\theta(d\theta/dz)$ ). Hence, we hypothesise that the changes in background temperature structure due to the CO2 radiative effects alter the propagation of Rossby waves, particularly in the upper stratosphere, and this leads to the changes in EPFD shown in Figures 7 and 8."

2. The EP flux divergence plots seem to be inconsistent with the changes in stream function. The downward control principle indicates that the latitudinal distribution of Psi\* anomalies should be roughly consistent with those of EP flux divergence anomalies. However, the EP flux divergence anomalies seems to locate much poleward than Psi\* anomalies, especially in DJF over the Southern Hemisphere (Fig. 6 vs. Fig. S1, Fig. 7 vs. Fig. S2). Based on the argument that stronger subtropical jets following warming allow more waves penetrate into the stratosphere, one would expect the anomalous wave dissipation to occur at the subtropics. This also seems to disagree with the pattern shown in Fig. 6 and 7, where maximum wave dissipation occurs around 50-60 degrees.

The torque exerted on the zonal flow and the associated w\* anomalies is proportional to EPFD  $\times$  cos(lat) (Haynes et al., 1991); this explains the apparent difference in pattern of EPFD and psi\*. When the EPFD is multiplied by cos(lat) the patterns more strongly resemble one another (see Figure R1 below), as expected. Furthermore, this weighted pattern of EPFD anomalies compares closely with the distribution of wave forcing found by Shepherd and McLandress, (2011) who showed in detail the role of changing Rossby wave critical lines in the subtropical lower stratosphere. We have replaced Figures 6 and 7 of the main text with the below figures which show the cosine(latitude) weighted EPFD anomalies. We have also replotted the parametrized wave forcing supplementary figures using the same scaling.

Figure R1: (left) As in Figure 6 of the main text, showing DJF EPFD anomalies for the four experiments, but multiplied by cosine of latitude. (right) As in Figure 7 of main text, showing JJA EPFD anomalies.

3. Model bias in climatology. The model simulated turn-around latitude in piControl climatology seems to be too poleward compared to reanalysis or other models (e.g., Fig. 1 in Hardiman et al. (2014)). At around 20 hPa, there are multiple turn-around latitude in the Northern Hemisphere, indicating more than one cell of the circulation, which seems unreasonable. Such bias in the mean circulation structure reflects bias in the waves forcing and/or other background condition. Would such model bias affect the model's ability to simulate the circulation changes in response to CO2 increase?

To further investigate this feature in the turnaround latitudes, Figure R2 below shows the w\* anomalies and turnaround latitudes by season. This shows the feature in the NH midlatitude middle stratosphere is a feature of the DJF season. There is a similar but less pronounced feature in the SH in JJA. We agree that, like any model, HadGEM3 contains biases which may affect the specific results of the study. However, we emphasise that the overall simulated changes in residual circulation correspond very well with other studies analysing large forced changes in the BDC in a range of climate models (e.g., Shepherd and McLandress, 2011; Hardiman et al., 2014; Lin et al., 2015). We have added a sentence at the end of the conclusions caveating that the results are from one model and that climatological biases might affect its simulation of the forced response to CO2: "The model contains mean biases that could affect some of the details of the responses described here."

---

## Author Comment (AC3) · 30 Mar 2020

**Author response to referee comments on Chrysanthou et al. (2020) "Decomposing the response of the stratospheric Brewer-Dobson circulation to an abrupt quadrupling in CO₂" submitted to Weather and Climate Dynamics Discussions**

**Reply to Anonymous Referee #3**

Changes in the Brewer-Dobson circulation (BDC) due to increased CO2 levels are studied by distinguishing the response to CO2 changes in the atmosphere only, globally uniform changes in SSTs, and SST pattern changes. The former corresponds to the rapid-adjustment of the climate system when CO2 levels are increased abruptly. The latter two correspond to long-term changes due to the longer time scales of the oceanic response. These effects are studied consistently by using a single state-of-the-art climate model (HadGEM3-A). The BDC generally increases in strength due to increased CO2. The authors find that in the lower stratosphere the majority of this BDC strengthening can be attributed to globally uniform SST increase. In the upper stratosphere the changes due to rapid adjustment are of similar magnitude. The authors furthermore estimate a linear sensitivity of the change in BDC strength as a function of global surface warming of roughly 9 %/K in the lower stratosphere and 6 %/K in the upper stratosphere. Overall, the paper is well-written and the results are straightforward. I have a few general comments that I hope will help the authors to sharpen their discussion and to better put the work into broader context. Other than that I only have minor comments; once these comments have been taken into account this manuscript should be publishable.

Thank you for your positive comments and suggestions to sharpen the discussion and enhance the readability of the study. We reply to the specific points raised below in red.

**General comments:**

SST pattern changes and ENSO: there are frequent remarks about the resulting BDC changes from the SST pattern changes to be similar to ENSO-induced anomalies. However, in the discussion section (line 403) the authors remark that "the SST pattern imposed here is very different from a canonical ENSO SST pattern". If that is the case, isn't it surprising then that the BDC changes due to the SST pattern changes look similar to those due to ENSO? To me this calls for corresponding discussion/elaborations somewhere in the manuscript.

Thanks for the comment. The issue of the pattern was also raised by reviewer 1. On reflection this description was an oversimplification. ENSO SST anomalies are confined to the tropical Pacific, whereas the 4xCO₂ SST pattern shows features globally (by construction). This includes relatively higher SST across the tropical oceans and North Pacific and relatively cooler SSTs in the Southern Ocean. We have amended the text in the Methods to provide a more nuanced discussion about the features of the 4xCO₂ SST pattern (lines 163-165). We still discuss the BDC response to ENSO but only as a point of comparison in terms of the magnitude of its effect (tracked changes manuscript lines 461-462).

Shallow versus deep branch changes: it seems that the authors interpret changes in upwelling strength through 70 hPa as representative of the shallow BDC branch, whereas those at 10 hPa as representative of the deep branch. Although it is certainly true that there isn't a clear vertical level where the shallow branch stops and the deep branch takes over, perhaps a useful distinguishing factor is related to the tropical pipe concept (much less meridional exchange at pressure levels within the tropical pipe than below). The shallow branch could then be interpreted as being primarily confined to that part of the BDC that involves strong meridional dispersion below the bottom of the tropical pipe. My recollection is that this level (bottom of tropical pipe) is close to 70 hPa, although this may vary from model to model. By that argument the upwelling through 70 hPa is more a measure of the deep rather than the shallow BDC branch and neither

of the quoted upwelling changes correspond to the shallow branch strength. I admit that all of this may be a bit philosophical, but the authors may wish to include a bit of discussion on this point. This isn't an issue when simply referring to 70 vs. 10 hPa without the connotation of shallow vs. deep branch. But even in that case, one wonders about BDC changes in the lower half of the stratosphere (assuming a global mean tropopause pressure somewhere around 150 hPa, roughly half of the stratosphere is located below 70 hPa)

We agree there is no accepted pressure level to delineate the different branches of the BDC, that different levels have been used throughout the literature, and that these levels likely vary in models too. Our choice of levels to focus on was partly motivated by the comparison with earlier model intercomparisons (e.g., Fig 4.10 SPARC, 2010), which tended to analyse the BDC at 70 hPa and 10 hPa. For clarity, we have amended the text to always refer to the specific pressure levels and/or to the lower and upper stratosphere rather than to the shallow and deep branches. Based on the reviewer's points about the mass flux in the lower stratosphere, we have also amended our cross-section plots to show levels down to 150 hPa.

Seasonal versus annual means: residual circulation changes are shown in terms of annual means in the main manuscript, whereas those of EP flux divergence are shown in terms of seasonal means. Line 308 presents a specific argument in favour of seasonal means. I didn't understand why this argument should apply to the wave forcing but not the resulting residual circulation, hence why seasonal means were delegated to the supplement in the case of the residual circulation? Please clarify somewhere.

Following this comment, we have moved the seasonal mean psi* Figures S1 and S2 to the main text and moved the annual mean psi* in Fig. 5 to the supplement. This part of the manuscript was rewritten to reflect the above changes.

**Minor comments:**

line 47: not sure I can follow the argument here - why couldn't the wave forcing around the turnaround latitudes change if there was a change in wave activity from the troposphere?

This sentence was indeed misleading. We meant that the wave forcing *change* needs to take place near the turnaround latitudes (TL) to directly affect the BDC. Poleward of the TL and within the deep tropics it will only lead to a latitudinal re-distribution of the downwelling or upwelling respectively (Shepherd and McLandress, 2011). We have updated the text to clarify this point.

line 51: the Randel and Held reference is appropriate for the connection of wind pattern and critical levels, but I don't think these authors talked about an upward movement of critical levels due to climate change; so the placement of this reference may be misleading

Indeed, the placement of this reference was misleading, so we have now removed it. We added a new sentence to better put into context the observed link between the wind patterns and the critical levels of wave breaking associated with an accelerated BDC under climate change.

line 81: GEOS-CCM model: the acronym "CCM" already contains "model"

Thanks for spotting this, we have now removed the word "model".

line 88: here and at other place: "warmer SSTs" should be something like "higher SSTs" (or "warmer sea surfaces")

Following your suggestion, we have changed the warmer SSTs to "higher" SSTs.

line 106: "three distinct effects" – would be good to briefly remind reader about the effects

This suggestion surely improves the readability in this part of the manuscript, so we briefly list these effects. Thanks!

line 144: is the runtime for the 4xCO2 experiments long enough to call the final state "quasi-equilibrium"? I could imagine that there's still drift due to ocean response, even after 150 years.

The reviewer is correct that the climate has not fully equilibrated after 150 years. We have removed the phrase "quasi-equilibrium" and replaced it with "centennial".

line 147/148: please add comment about the 3.4 K warming, especially comparing it to the equilibrium response to 4xCO2 (which should be double the climate sensitivity if I understand correctly, so the 3.4 K value seems small)

This value is not comparable to the equilibrium climate sensitivity (ECS), since this is the global surface temperature (land+ocean) and we consider only the global mean SST change. Since the land warms more than the ocean, we expect our imposed global SST change to be smaller than 2xECS. Furthermore, our warming is calculated over years 101-150 whereas ECS extrapolates to equilibrium which takes millennia to reach (Rugenstein et al., 2020). We have added a comment in the Methods on this point: "Note the global mean SST is smaller than the global mean surface temperature change in the abrupt-4xCO2 experiment because land areas warm more than the ocean (e.g., Joshi and Gregory, 2008)."

line 175 and following: please clarify use of vertical coordinate; your model runs in height coordinates (not log-p height), but the TEM diagnostics are formulated in log-p height – was this done by first interpolating the data?

Thank you for spotting this, the equations were incorrectly written in the log-p coordinate system although the MetUM uses the primitive equations with the log-pressure $z = -H \ln(p/ps)$ coordinate. We did not perform any interpolation; we used the direct model output which was calculated based on the equations 3.5.1a and 3.5.1b from Andrews et al., (1987).

line 181: I assume that Eq. 2 is only integrated to the respective vertical level of interest (so that Psi* is still function of log-p height), not all the way to the surface (unless for lowest level)? Also, your definitions in Eqs. 1 and 2 are circular: Eq. 1 requires knowledge of Psi* and Eq. 2 requires knowledge of v* ... even though these are standard diagnostics, it would be more helpful to define the residual streamfunction first based on the vertically integrated v and the heat flux contribution; then define v* and w* (or, alternatively, define v* and w* in terms of v and w + heat flux contribution; then use Eq. 2 for Psi*).

For the first part of the question, yes for each level, we integrate from the top of the model to that particular level. Following up from your previous comment, we now define the residual circulation equations as shown in Andrews et al., (1987), changing our equation 1 and keeping equation 2 as it is in the manuscript. We have also updated the text to reflect these changes.

Eq. 4: the integral is missing a "dz"

Thanks for spotting this, we have now added it.

line 244: please make more definite statements about the direction of changes of these QBO characteristics (or omit the comment altogether)

We feel it would be a separate study in its own right and that to offer enough detail to satisfactorily explain the QBO changes would detract from the main focus of the paper. We have therefore amended the text: "This is likely related to changes to the QBO properties under climate change, which have been noted in other idealised GCM experiments (e.g. Kawatani et al., 2011), though a detailed investigation of the QBO is beyond the scope of this study".

line 259: this is a good example where you make a reference to ENSO-like SST perturbations, but fall short in discussing how your SST pattern changes actually do correspond to ENSO (or not)

Thanks for pointing this out. The issue of the pattern was also raised by reviewer 1. While the SST pattern shows an El Niño-like warming across the equatorial Pacific, the pattern shows other pronounced features such as relatively warmer SSTs across all tropical ocean basins and the North Pacific and relatively cooler SSTs across the Southern Ocean. We have clarified the text on the SST pattern and its interpretation in relation to the local tropical Pacific anomalies vs. the global pattern.

line 263: this is a good example where I found your reference to the shallow BDC branch confusing – to me it doesn't really extend to 30 hPa

We have reworded the text to make this clearer.

lines 299/300: should this result perhaps be shown / referred to right away with the methods section? Also: it sounds a bit misleading to me to start the paragraph with "An important question" and then talk about results shown in the supplement – if they really are important, why aren't they shown in the main part of the paper?

We prefer to place this discussion here because it follows from the detailed discussion of the individual responses and would therefore be premature in the Methods. We have reworded the opening phrase of the paragraph from "An important question..." to "We lastly consider...".

line 330: SSW -> SST

Thank you for spotting this, we have now corrected it.

line 350: could you elaborate where this 20 % disagreement could come from?

The direct psi* calculation is derived from v* (see Equation 2). This was used because it was found to be less noisy than performing an equivalent integration of w* in latitude. However, in previous work we have noticed that, for model pressure level data, the psi* estimated from v* is generally larger than that from w* (see Fig. S2 of Dietmüller et al., 2018). We hypothesise that this is the source of the difference between the direct and downward control calculations, though we cannot explain its origin. We have added a reference to this in the text.

line 400: the statement is based on results from this paper, so I assume the reference to Lin et al. is meant to state that they found similar results? Please clarify

Thanks for spotting this, this was referring to the fact that the Lin et al., (2015) study had similar findings. We have slightly updated the text to better communicate this.

line 434: this value (~9 %/K) is exactly equal to the one you quote for your results, so the agreement is exact (or almost exact) and not just "relatively good" - am I missing something?

Thank you for pointing this out, as a matter of fact it is in exact agreement. We have now updated the text to reflect this.

line 675 (Fig. 3 caption): the u=0 lines are only critical lines for stationary waves - please clarify

Thank you for bringing this to our attention. We now clarify that the critical lines refer to stationary waves.

Fig. 4 and related discussions: visually, it doesn't look like the positive anomalies compensate the negative anomalies on a given pressure level (perhaps when scaled by surface area they do), but shouldn't they based on mass conservation? Did you check?

Yes, we have verified that the cos(latitude) area weighted w* anomalies on a given pressure surface produce a very small residual, e.g. at 70 hPa the area average value is -0.013 mm/s for the full experiment (annual mean), which is around 20 times smaller than the magnitude of anomalies at individual latitudes.

**References**

Andrews, D. G., Leovy, C. B., Holton, J. R. and Leovy, C. B.: Middle Atmosphere Dynamics, Academic press., 1987.

Dietmüller, S., Eichinger, R., Garny, H., Birner, T., Boenisch, H., Pitari, G., Mancini, E., Visioni, D., Stenke, A., Revell, L., Rozanov, E., Plummer, D. A., Scinocca, J., Jöckel, P., Oman, L., Deushi, M., Kiyotaka, S., Kinnison, D. E., Garcia, R., Morgenstern, O., Zeng, G., Stone, K. A. and Schofield, R.: Quantifying the effect of mixing on the mean age of air in CCMVal-2 and CCMI-1 models, Atmospheric Chemistry and Physics, 18(9), 6699–6720, doi:10.5194/acp-18-6699-2018, 2018.

Joshi, M. and Gregory, J.: Dependence of the land-sea contrast in surface climate response on the nature of the forcing, Geophysical Research Letters, 35(24), L24802, doi:10.1029/2008GL036234, 2008.

Kawatani, Y., Hamilton, K. and Watanabe, S.: The Quasi-Biennial Oscillation in a Double $CO_2$ Climate, Journal of the Atmospheric Sciences, 68(2), 265–283, doi:10.1175/2010JAS3623.1, 2011.

Lin, P., Ming, Y. and Ramaswamy, V.: Tropical climate change control of the lower stratospheric circulation, Geophysical Research Letters, 42(3), 941–948, doi:10.1002/2014GL062823, 2015.

Rugenstein, M., Bloch-Johnson, J., Gregory, J., Andrews, T., Mauritsen, T., Li, C., Frölicher, T. L., Paynter, D., Danabasoglu, G., Yang, S., Dufresne, J., Cao, L., Schmidt, G. A., Abe-Ouchi, A., Geoffroy, O. and Knutti, R.: Equilibrium Climate Sensitivity Estimated by Equilibrating Climate Models, Geophysical Research Letters, 47(4), 1–12, doi:10.1029/2019GL083898, 2020.

Shepherd, T. G. and McLandress, C.: A Robust Mechanism for Strengthening of the Brewer–Dobson Circulation in Response to Climate Change: Critical-Layer Control of Subtropical Wave Breaking, Journal of the Atmospheric Sciences, 68(4), 784–797, doi:10.1175/2010JAS3608.1, 2011.

SPARC: SPARC CCMVal Report on the Evaluation of Chemistry-Climate Models. V. Eyring, T. Shepherd and D. Waugh (Eds.), SPARC Report No. 5, WCRP-30/2010,WMO/TD-No.40 [online] Available from: http://www.sparc-climate.org/publications/sparc-reports/sparc-report-no5/, 2010.